



**Understanding summertime peroxyacetyl nitrate (PAN) formation and**
**its relation to aerosol pollution: Insights from high-resolution**
**measurements and modeling**
**Baoye Hu[1,3,4], Naihua Chen[1,6], Rui Li[7], Mingqiang Huang[1,3,4], Jinsheng Chen[2,5*], Youwei Hong[2,5],**
**Lingling Xu[2,5], Xiaolong Fan[2,5], Mengren Li[2,5], Lei Tong[2], Qiuping Zheng[8], Yuxiang Yang[6*]**
[1]College of Chemistry, Chemical Engineering and Environment, Minnan Normal University, Zhangzhou, China, 363000
[2]Center for Excellence in Regional Atmospheric Environment, Institute of Urban Environment, Chinese Academy of Sciences,
Xiamen 361021, China
[3]Fujian Provincial Key Laboratory of Modern Analytical Science and Separation Technology, Minnan Normal University,
Zhangzhou, China, 363000
[4]Fujian Province University Key Laboratory of Pollution Monitoring and Control, Minnan Normal University, Zhangzhou, China,

12 363000

[5]Fujian Key Laboratory of Atmospheric Ozone Pollution Prevention, Chinese Academy of Sciences, Xiamen 361021, China
[6]Pingtan Environmental Monitoring Center of Fujian, Pingtan 350400, China
[7]Key Laboratory of Geographic Information Science of the Ministry of Education, School of Geographic Science, East China
Normal University, Shanghai 200241, PR China
[8]Xiamen Key Laboratory of Straits Meteorology, Xiamen Meteorological Bureau, Xiamen 361012, China
*Correspondence to:* Jinsheng Chen (jschen@iue.ac.cn) & Yuxiang Yang (907460293@qq.com)
**Abstract:** Peroxyacetyl nitrate (PAN), a key indicator of photochemical pollution, is generated through a process similar to ozone
($O_3$), involving the photochemical reactions of specific volatile organic compounds (VOCs) in the presence of nitrogen oxides.
Notably, PAN has been observed at unexpectedly high concentrations (maximum: 3.04 ppb) during summertime that the daily
maximum values of PAN were better correlated to black carbon (BC) ($R^2$=0.85) than ozone ($O_3$) ($R^2$=0.75), suggesting that
summertime haze and photochemical pollution were deeply connected. We addressed the puzzle of summertime PAN formation
and its association with aerosol pollution under high ozone conditions by analyzing continuous high temporal resolution data
utilizing box modeling in conjunction with the master chemical mechanism (MCM). With an IOA value of 0.75, the MCM model
proves to be an ideal tool for investigating PAN photochemical formation. The model performed better during the clean period ($R^2$:
0.6782, slope K: 0.9097) than during the haze period ($R^2$: 0.4708, slope K: 0.7477). Through the machine learning method of
XGBoost, we found that the top three factors leading to simulation bias were $NH_3$, $NO_3$, and $PM_{2.5}$. Moreover, the net production



rate of PAN becomes negative with PAN constrained, suggesting the existence of an unknown compensatory mechanism. Both
RIR and EKMA analyses indicate that PAN formation in this region is VOC-controlled. Controlling emissions of VOCs,
particularly alkenes, $C_5H_8$, and aromatics, would mitigate PAN pollution. RIR results also show that during the clean period, PAN
is more sensitive to changes in various pollutants than during the haze period, underscoring the importance of deep emission
reductions. PAN promotes OH and $HO_2$ while inhibiting the formation of $O_3$, $RO_2$, NO, and $NO_2$. This study deepens our
comprehension of PAN photochemistry while also offering scientific insights for guiding future PAN pollution control strategies.

**Introduction**
Peroxyacetyl nitrate (PAN, $CH_3C(O)ONO_2$) is a significant secondary gaseous pollutant commonly present in photochemical
smog and poses risk to human health and plant growth, being 1-2 magnitudes more phytotoxic than ozone ($O_3$) (Yukihiro et al.,
2012; Taylor, 1969). Additionally, PAN's low aqueous solubility, minimal reactivity with hydroxyl radicals (OH), and slow
photolysis contribute to its capacity for long-range transport of nitrogen oxides (NOx) (Xu et al., 2018; Zhai et al., 2024; Marley
et al., 2007b). Therefore, its formation in polluted areas holds significant importance beyond local concerns. Similar to surface $O_3$,
PAN is produced during the oxidation of volatile organic compounds (VOCs) in the presence of NO$x$ (R1-R3). PAN is formed
when $NO_2$ reacts with peroxyacetyl (PA) radicals ($CH_3C(O)OO\bullet$) (R2), but the presence of NO consumes PA radicals, inhibiting
PAN production (R3), which creates a comparable dependence of PAN and $O_3$ on NO and $NO_2$ levels (Xu et al., 2021). Unlike $O_3$,
however, PAN is influenced by only a limited number of oxygenated VOCs (OVOCs) that generate PA radicals. These OVOCs,
which are second-generation precursors of PAN, include acetaldehyde ($CH_3CHO$), acetone ($CH_3C(O)CH_3$), methylglyoxal
(MGLY, $CH_3C(O)CHO$), methyl vinyl ketone (MVK, $CH_2CHC(O)CH_3$), methyl ethyl ketone (MEK, $CH_3C(O)CH_2CH_3$),
methacrolein (MACR, $CH_2C(CH_3)CHO$), and biacetyl ($CH_3C(O)C(O)CH_3$). These compounds are typically formed from the
oxidation of alkenes, aromatics, and isoprene, which are the first-generation precursors of PAN (Xue et al., 2014; Zhang et al.,
2015). Identifying the dominant precursors is crucial for managing PAN pollution effectively. In the troposphere, thermal
decomposition (R4) is the primary process responsible for PAN loss (Xu et al., 2021).
$VOC_s + hv/OH/O_3\,/NO_3 \xrightarrow{O_3} CH_3C(O)OO \cdot + products$                                     R1
$CH_3C(O)OO \cdot + NO_2 \xrightarrow{k_2} PAN$                                                              R2
$CH_3C(O)OO \cdot + NO \xrightarrow{k_3} CH_3 \cdot + NO_2 + CO_2$                                           R3
$PAN \xrightarrow{k_4} CH_3C(O)OO \cdot + NO_2$                                                             R4
In recent years, wintertime photochemical air pollution has increasingly garnered attention. At this time, the concentration of $O_3$ is
low due to the strong titration of NO, while the concentration of aerosol is high, and it is found that aerosol promotes PAN
generation. Surprisingly high concentrations of OH radical, particularly under hazy conditions, have been observed and are largely



attributed to HONO photolysis. Winter photochemical and haze pollution often exacerbate each other, with photochemical trace
gases supplying both oxidants and precursors for aerosol formation, and aerosols acting as mediums for heterogeneous reactions
that produce key oxidants such as HONO, $H_2O_2$, and OH radicals (Xu et al., 2021). The OH produced by HONO photolysis can
partially replace the UV action to promote PAN formation in winter in southeast coastal area of China when particulate matter is
high ($\geq 35\mu g \cdot m^{-3}$) (Hu et al., 2020). Zhang et al. (2020) found the potential HONO sources significantly improved the PAN
simulations in wintertime heavy haze events with high concentrations of PAN. High concentrations of PAN are a consequence of
the increased levels of precursors and HONO observed during haze episodes (Liu et al., 2018). In conclusion, most previous
studies have studied the effect of aerosol on PAN generation in winter. Further research on PAN should determine whether
particulates significantly contribute to its formation during warmer seasons with elevated ozone concentrations (Xu et al., 2021).
In Eastern China, photochemical air pollution often involves high concentrations of both $O_3$ and PAN, a persistent issue during the
warm season (April-September) for many years (Lu et al., 2020). The characteristics and formation pathways of PAN during
summer have been increasingly studied in regions such as the North China Plain (NCP), the Yangtze River Delta, the Pearl River
Delta, and southwestern China. These studies have generally shown consistent diurnal patterns and strong correlations between
PAN and $O_3$, identifying acetaldehyde—primarily derived from the degradation of aromatics and alkenes—as the key direct
precursor of PAN in the summer. However, there has been limited research on the formation of peroxyacetyl nitrate (PAN) and its
relationship with aerosol pollution during the summertime.
Ximen is one of the fastest urbanizing regions in the southeast China and is also one of the cities with the best air quality in China,
where the air quality could represent the future of other Chinese urban regions. Xiamen is also often affected by typhoons and
subtropical weather forms in summer. The West Pacific Subtropical High (WPSH) creates weather conditions that promote the
formation and accumulation of photochemical pollutants and particulate matter (Wu et al., 2019). This setting provides an ideal
"laboratory" for investigating the complexities of summertime PAN formation and its relationship with aerosol pollution under
high ozone concentrations. In this study, continuous measurements of trace gases, substances related to aerosols, photolysis rate
constants and meteorological parameters were performed at a suburban site in Xiamen from July 10th to July 31st, 2018. Firstly,
we provide an overview of pollutant concentrations, meteorological parameters, and weather conditions during the observation
period. Secondly, we simulate PAN concentration with the aid of box modeling combined with master chemical mechanism
(MCM). Using a machine learning-XGBoost looks for the key factors that affect the model's simulation result and clarified the
mechanisms linking haze pollution to photochemical air pollution, as indicated by PAN and $O_3$. Thirdly, the study identified the
main precursors and oxidants responsible for summertime PAN production in Xiamen and evaluated the influence of PAN on local
atmospheric oxidation capacity. This study further emphasized the interplay between haze and photochemical air pollution and
highlighted significant implications for future research.



## 2 Methodology

### 2.1 Field observations

Trace gases (including PAN, $O_3$, HONO, $HNO_3$, HCl, $NH_3$, VOCs, NO$x$, CO, and $SO_2$), substances related to aerosols (including BC, $PM_1$, $PM_{2.5}$, $PM_{10}$, $SO_4^{2-}$, $NO_3^-$, $NH_4^+$, Cl$^-$, Na$^+$, K$^+$, Ca$^{2+}$, Mg$^{2+}$), photolysis rate constants (including JO$^1$D, $JNO_2$, JHONO, JHCHO_M, JHCHO_R, JNO$_3$_M, JNO$_3$_R, JH$_2$O$_2$), and meteorological parameters (including temperature, relative humidity, atmospheric pressure, wind speed, and wind direction) were continuously measured at an suburban site in Xiamen from July 10th to July 31st, 2018. All instruments were placed inside an air-conditioned container situated on the rooftop of a 20-story building at the Institute of Urban Environment, Chinese Academy of Sciences (IUE: 118.06°E, 24.61°N) (Fig. S1(a)). When southerly winds prevailed, Xiamen Island, characterized by dense population and traffic congestion, was located upwind of the IUE (Fig. S1(b)). The IUE supersite is surrounded by Xinglin Bay, several universities and institutes, and major roadways with heavy traffic, such as Jimei Road (< 200 m), Shenhai Expressway (870 m), and Xiasha Expressway (2300 m) (Fig. S1(c)).

PAN measurements were conducted using a PANs-1000 analyzer (Focused Photonics Inc., Hangzhou, China), which features an automated system consists of a gas chromatograph, an electron capture detector, and a calibration unit. The analyzer provided PAN readings every 5 minutes, with a detection limit of 50 ppt. The uncertainty and precision of the PAN measurements were ±10% and 3%, respectively. The PAN standard gas was produced through the reaction of acetone and NO under UV light. Calibration procedures included monthly multi-point calibrations and weekly single-point calibrations. Detailed information about the PAN detection system and calibration can be found in previous studies (Hu et al., 2020; Liu et al., 2022). HONO measurements were conducted using a customized Incoherent BroadBand Cavity Enhanced Absorption Spectroscopy (IBBCEAS) system developed by the Anhui Institute of Optics and Fine Mechanics (AIOFM), Chinese Academy of Sciences. The HONO detection limit was 100 ppt, with a time resolution of 1 minute. The measurement principle and calibration method of IBBCEAS can be found in the previous literature (Hu et al., 2022; Duan et al., 2018; Hu et al., 2024). The concentrations of inorganic components in $PM_{2.5}$ aerosols (including $SO_4^{2-}$, $NO_3^-$, $NH_4^+$, Cl$^-$, Na$^+$, K$^+$, Ca$^{2+}$, Mg$^{2+}$), as well as the concentrations of gases such as $NH_3$, HCl, and $HNO_3$ were analyzed using a Monitor for AeRosols and Gases in ambient Air (MARGA, Model ADI 2080, Applikon Analytical B.V., the Netherlands) (Hu et al., 2022). The criteria air pollutants $O_3$, NO$x$, CO, and $SO_2$ were measured using different methods: ultraviolet (UV) absorption for $O_3$ (TEI model 49i), chemiluminescence with a molybdenum converter for NO$x$ (TEI model 42i), non-dispersive infrared for CO (TEI model 48i), and pulsed UV fluorescence for $SO_2$ (TEI model 43i). A tapered element oscillating microbalance (TEOM1405, Thermo Scientific Corp., MA, USA) was used to continuously measure the mass concentrations of $PM_1$, $PM_{2.5}$, and $PM_{10}$ online. A photolysis spectrometer (PFS-100, Focused Photonics Inc., Hangzhou, China) was employed to measure the photolysis rate constants. An ultrasonic atmospherium (150WX, Airmar, USA) was used to measure meteorological parameters.



**2.2 Box modeling**

This study employed a box model framework utilizing the Master Chemical Mechanism (MCMv3.3.1, https://mcm.york.ac.uk/MCM/home.htt) to investigate sensitivity and mechanisms of PAN formation. The model constraints were derived from observations of trace gases and meteorological parameters, which were averaged to 1-hour intervals. The reliability of model simulation results is often assessed using the index of agreement (IOA), which ranges from 0 to 1, with a higher IOA signifying greater alignment between observed and simulated values. Note that the model simulation values at this time are not constrained by PAN. For specific formulas, please refer to the supplementary information (Eq. S1). Other formulas, including PAN production rates (P(PAN)), net production of PAN (Net (PAN)), and the relative incremental reactivity (RIR), are provided in the supplementary information (Eq. S2- Eq. S4).

The MCM simulates the nonlinear interaction between PAN and its precursors by altering the VOCs-to-NOx ratio across multiple scenarios, while keeping all other parameters fixed. In this study, a 20% step size was applied, reducing VOCs and NOx from 200% down to 0% to construct a scenario matrix. A total of 121 scenarios were generated to model the PAN production rate. The scenario representing the average VOCs and NOx mixing ratio during the sampling period was designated as the base case, with the remaining 120 scenarios created by systematically adjusting the VOC-to-NOx ratio. The output from these 121 simulations was used to construct isopleth diagrams depicting the relationship between VOCs, NOx, and PAN.

**2.3 Machine Learning Model**

To identify the key factors influencing the performance of the model simulation, the Machine Learning (ML) model was applied to establish the prediction model of bias between simulation of OBM and observation. XGBoost is a supervised boosting algorithm that reduces the risk of over-fitting, captures the nonlinear relationships among predictor variables, and solves numerous data science problems in a rapid and accurate way (Li et al., 2024). It has demonstrated high performance in $O_3$ studies in over China. As compared to other bagging tree models like random forest, XGBoost can handle more complex data while consuming fewer computing resources. To further improve the interpretability of the ML model, the feature importance of independent input variables in the XGBoost model is quantified using the Shaply Additive explanation (SHAP) approach. The SHAP calculates a value that represents the contribution of each feature to the model's outcome, which has been successfully applied in atmospheric environmental studies. When the model was being adjusted, 90% of the data was used as the training set, and 10% of the data was used as the test set. The hyperparameters were tuned using grid search and cross-validation method. Specifically, for a single hyperparameter, grid search was used to obtain its more appropriate value range, and for the combinations of hyperparameters, the whole training set was split into ten folds and then run a grid search over pre-adjusted combinations of hyperparameters by training nine folds and predicting on the one fold in cross-validation procedure. For key hyperparameters of XGBoost model, the number of trees was 100, learning rate was 0.1, max depth was 6. The model was trained and tested on hourly data during the whole observation and the established model was examined by coefficient of determination ($R^2$) value, the root-mean-squared




error (RMSE) and mean absolute error (MAE). The formulas of RMSE and MAE are provided in the supplementary information
(Eq. S5 & Eq. S6). The performance of both models is illustrated in Fig. S3. The R², MAE, and RMSE for the training set are
0.9037, 0.08, and 0.12, respectively, while the corresponding values for the test set are 0.7664, 0.10, and 0.14, respectively. These
statistical metrics indicate that the XGBoost model is promising for further analysis.

**3. Results and discussion**
**3.1 Overview of observation**
The measured data of PAN, related trace gases and meteorological parameters at IUE over 10-31 July 2018 are documented in Fig.
1. Combine with the synoptic situation (Fig. S4), the 8th typhoon of 2018, Typhoon Maria, made landfall on the morning of the
11th in Huangqi Peninsula, Lianjiang County, Fujian. Due to the influence of the typhoon's outer spiral rain bands, there was
moderate to heavy rain on the 11th. Correspondingly, there was a noticeable decrease in ultraviolet radiation and the temperatures.
Starting from the 12th, a Western Pacific subtropical high (WPSH) strengthened and extended westward, exerting control over
Xiamen city. In the lower atmosphere, it was influenced by the eastward flow, resulting in predominantly cloudy weather. From
the 16th to the 18th, the area was affected by the outer periphery of Typhoon Shan Shen, which formed on the 17th in the
northeastern part of the South China Sea and moved westward, making landfall along the coast of Wancheng Town, Wanning City,
Hainan Province in the early hours of the 18th. During this period, the city experienced strong winds with gusts reaching 5 to 6 on
the Beaufort scale in the urban areas. At the same time, the concentration of various pollutants reached their lowest levels, and the
daily variation patterns were less pronounced. From the 20th to the 21st, Xiamen City experienced the influence of the peripheral
descending airflow associated with Typhoon Ampil (which formed in the northwest Pacific Ocean around 8:00 P.M. on the 18th
and moved northwest, making landfall along the coast of Chongming Island, Shanghai, around noon on the 22nd). During this
period, there were fewer clouds and higher temperatures. From the 22nd to the 24th, the city was successively affected by the
outer periphery of Typhoon Ampil and a tropical low-pressure system, resulting in occasional showers or thunderstorms. From the
25th to the 31st, a WPSH once again strengthened and controlled Xiamen City. As a result, Xiamen city experienced stable
meteorological conditions, with light winds (ws = 1.04 m/s), persistently high temperatures (daily maximum average of 37.82 °C),
and high relative humidity (daily maximum average of 81.65%). These factors created an environment that favored the buildup of
particulate matter and enhanced the photochemical formation of $O_3$ and PAN (Wu et al., 2019). The daily maximum average of
$PM_{2.5}$, $O_3$ and PAN were 49.26 μg.m$^{-3}$, 93.62 ppb, and 1.37 ppb, respectively.
The phenomenon of simultaneous high levels of photochemical and particulate matter appears. Throughout the 22-days campaign,
12 days (including July 11th, 13th, 21st to 23rd, and 25th to 31st) were observed with 1 h concentrations of $PM_{2.5}$ exceeding 35
μg·m$^{-3}$; 13 days (including July 11th, 13th, 15th, 20th to 23rd, and 26th to 31st) were observed with 5-min concentrations of PAN
exceeding 1 ppb. The maximum concentration was recorded at 3.04 ppb (5-min data) at 11:09 local time of 13 July 2018. This





concentration of PAN is comparable to the levels recorded at downwind of Guangzhou, southern China (3.9 ppb) (Wang et al.,
2010), 2.51 ppb in Nashville, U.S (Roberts et al., 2002). However, this value was significantly lower than heavily polluted areas in
northern China in the summer, such as Beijing (9.34 ppb, (Xue et al., 2014)), Lanzhou (9.12 ppb, (Zhang et al., 2009)), and Jinan
(13.47 ppb, (Liu et al., 2018)). This is likely because the higher summer temperatures in the southeastern coastal region are
conducive to the thermal decomposition of PAN, and the precursor concentration of PAN is significantly lower than in the
northern region. Throughout the observation period, the variations in $O_3$ and PAN were almost identical, but the maximum
concentration of $O_3$ occurred at 3:00 p.m. on July 29[th] (114.12 ppb). The correlation between the daily maximum values of PAN
and BC is the strongest (R=0.85), followed by $O_3$ (R=0.75), suggesting that summertime haze and photochemical pollution were
deeply connected.

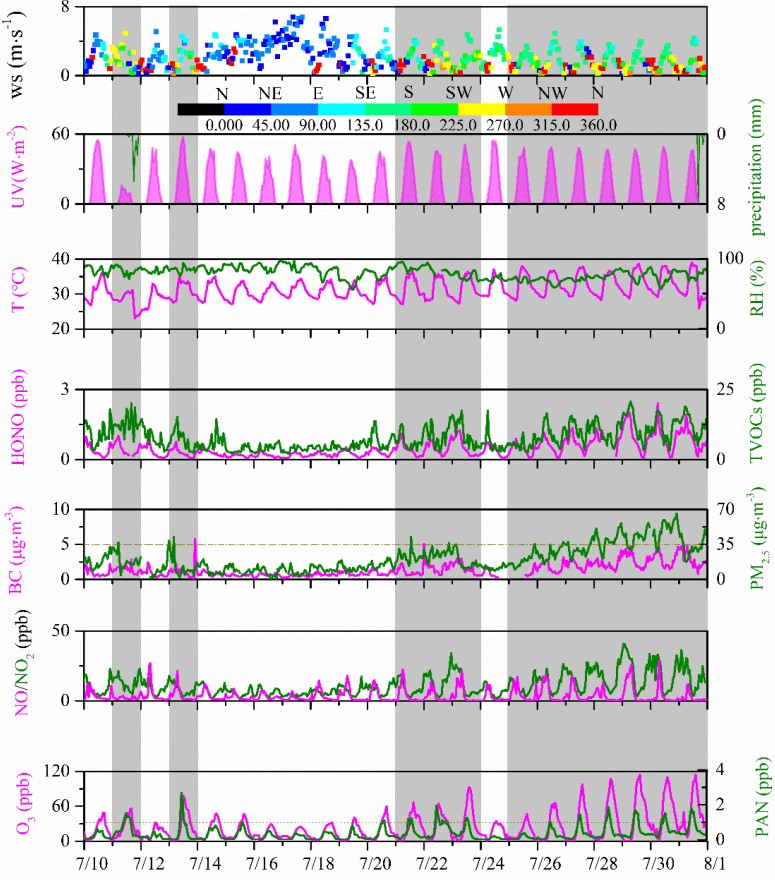


**Figure 1.** Time series of trace gases and meteorological parameters observed at IUE during 10-31 July 2018. The gray shading
represents days when the $PM_{2.5}$ hourly daily maximum value exceeded 35 $\mu g \cdot m^{-3}$.

We categorize it as "haze" and "clean" based on whether the $PM_{2.5}$ hourly daily maximum value is greater than 35 $\mu g \cdot m^{-3}$.



Specifically, "haze" includes July 11th, 13th, 21st to 23rd, and 25th to 31$^{st}$, while other days are categorized as 'clean'. To provide
a quantitative perspective, the statistics for PAN and associated species were calculated and compiled in Table 1. PM$_{2.5}$
concentrations during the haze period were significantly higher than during the clean period, being 2.49 times that of the clean
period. There was no significant difference of UV levels between clean and haze periods, while temperatures in the haze phase
were notably higher than those in the clean phase. Therefore, without considering precursors, PAN concentrations should be
lower during the haze phase due to higher thermal decomposition. In fact, PAN concentrations during the haze period were 2.35
times higher than those during the clean period. During the haze period, ozone concentrations were also significantly higher than
during the clean period, being 2.04 times that of the clean period. These observations indicate that the atmospheric oxidation
capacity is relatively strong during the haze period. Similar to PAN, HONO also exhibits higher concentrations during the haze
phase (approximately 2.33 times that of clean conditions), which is consistent with current research findings that particles
promote the generation of HONO (Ye et al., 2017). NO also experienced an increase from clean (3.28 ppb) to hazy (4.30 ppb)
conditions, albeit less prominently than NO$_2$ (from 7.21 to 14.55 ppb). This observation further underscores that, during hazy
periods, the atmosphere demonstrates heightened oxidizing potential, facilitating the conversion of NO to NO$_2$. While the
increased NO levels on hazy days reduced PA radicals and hindered PAN formation, this effect was offset by the concurrently
higher concentrations of PAN precursors (NO$_2$ and VOCs) during those days. The TVOCs have increased to some extent, but in
hazy conditions, they are only 1.34 times that of clean conditions. This is also because the strong oxidizing conditions during
haze periods convert VOCs into secondary pollutants, such as O$_3$ and PAN. The TVOC levels at this site are comparable to that
at a rural site in a coastal city-Qingdao (7.6 ppb), significantly lower than inland sites (such as Wuhan (30.2, (Liu et al., 2021a))
and Chengdu (28.0 ppb, (Yang et al., 2020))) or economically more developed coastal cities (such as Shanghai (25.3 ppb, (Zhu et
al., 2020)) and Hong Kong (26.9, (Wang et al., 2018))), and significantly higher than regional background locations like Mt.
Wuyi (4.7 ppb, (Hong et al., 2019)), Mt. Waliguan (2.6 ppb, (Xue et al., 2013)), and Mt. Nanling (4.7 ppb, (Wang et al., 2023)).
The isoprene level during haze period was significantly higher than that during clean period probably due to haze period with
higher temperature (Wang et al., 2023). The wind speed was very low during both the clean and haze periods, especially during
the haze period with only 1.12 m·s$^{-1}$. The relative humidity was high during both periods, and there was no significant difference
between the clean and haze periods.
**Table 1.** Descriptive statistics of major trace gases (ppb), particulate matter (µg·m$^{-3}$) and meteorological parameters during 10-31
July 2018.

| Species | Clean (mean ± SD) | Haze (mean ± SD) |
|---------|-------------------|------------------|
| PAN | 0.20 ± 0.23 | 0.47 ± 0.46*** |
| O$_3$ | 16.07 ± 12.73 | 32.79 ± 29.73*** |
| HONO | 0.27 ± 0.18 | 0.63 ± 0.43*** |
| NO | 3.28 ± 4.03 | 4.30 ± 8.39*** |





| | | |
|---|---|---|
| NO$_2$ | 7.21 ± 3.87 | 14.55 ± 8.89*** |
| TVOCs | 6.13 ± 1.73 | 8.19 ± 2.55*** |
| C$_5$H$_8$ | 0.13 ± 0.04 | 0.17 ± 0.05*** |
| PM$_1$ | 10.13 ± 3.91 | 24.36 ± 10.77*** |
| PM$_{2.5}$ | 11.21 ± 5.33 | 27.93 ± 13.16*** |
| PM$_{10}$ | 24.26 ± 9.45 | 47.28 ± 20.63*** |
| UV (W·m$^{-2}$) | 14.29 ± 17.38 | 13.18 ± 17.40 |
| T (°C) | 30.68 ± 2.39 | 31.92 ± 3.36*** |
| RH (%) | 81.94 ± 8.60 | 77.18 ± 8.22 |
| WS (m·s$^{-1}$) | 1.64 ± 0.69 | 1.12 ± 0.61* |

Note: *, **, and *** indicate that they passed the significance test at 0.05, 0.01 and 0.001 levels, respectively.

The average diurnal patterns of PAN and related variables have been averaged separately for clean and hazy conditions. The
daily variation of PAN exhibits a clear unimodal pattern, with concentrations starting to rise after sunrise and decreasing after
12:00 caused by thermal decomposition of PAN at high temperatures. The peak occurring at noon indicates that PAN primarily
originates locally, as a delay of about 1-2 hours would be expected if it were influenced by transportation (Liu et al., 2024). The
daytime increment was much larger for hazy condition (1.17 ppb) than for clean condition (0.52 ppb), indicating stronger
photochemical production of PAN for hazy condition. The daily variation pattern of ozone is similar to PAN, except that ozone
reaches its peak relatively later compared to PAN, with the peak occurring at 16:00 during the clean phase and 14:00 during the
haze phase. Although PAN and O$_3$ are both products of photochemical reactions involving NO$x$ and VOCs, their production
efficiencies differ. PAN is specifically formed from VOCs that are precursors to the acetyl radical (CH$_3$CO), whereas O$_3$ can be
produced from the oxidation of any VOCs. Analyzing the correlation between PAN and O$_3$ can offer insights into their respective
photochemical production efficiencies. As shown in Fig. S5, the positive correlation between the daily maximum values of PAN
and O$_3$ for clean condition (R$^2$=0.6701) was better than that for hazy condition (R$^2$=0.1504). The slopes of the linear regression
were 0.021 ppb/ppb for clean conditions and 0.009 for hazy conditions. This indicates that, on average, approximately 2.1 ppb of
PAN could be produced for each 100 ppb of O$_3$ formed under clean conditions, and about 0.9 ppb of PAN for each 100 ppb of O$_3$
under hazy conditions in the air masses reaching IUE. The slope of linear regression for clean condition is comparable to those
determined in Hongkong (0.028, (Xu et al., 2015)), Mexico (0.02, (Marley et al., 2007a)), and Nashville (0.025, (Roberts et al.,
2002)). The lower efficiency of PAN production relative to O$_3$ indicates that PAN precursors represent only a small portion of the
total VOCs, especially during hazy conditions. Additionally, the high temperatures in the southeast coastal region likely
contribute to the lower production efficiency of PAN. This result is consistent with the result that RIR during the cleaning period
is higher than that during the haze period. As shown in Fig. S6, in the clean period, the correlation between PAN and O$_3$ is the
strongest (R$^2$=0.7042), indicating that O$_3$ and PAN are both photochemical end products during clean periods. In contrast, during
hazy periods, the correlation between PAN and O$_3$×JO$^1$D is the strongest (R$^2$=0.6597), suggesting that O$_3$ plays a more



significant role in promoting PAN formation through photolysis to generate OH during hazy periods.
Unlike the daily variation patterns of PAN and $O_3$, HONO exhibits a swift concentration decrease after sunrise in both clean and
hazy conditions, undergoing photolytic conversion into OH radicals. Subsequently, in clean conditions, HONO starts to increase
in concentration after sunset. In hazy conditions, however, the increase begins from 16:00 LT and not after sunrise. This suggests
a robust daytime net production or transport of HONO, where the rates surpass those of HONO photolysis and other sinks in the
afternoon in hazy conditions. The NO levels reach their peak at 7:00 during the morning rush hour, reflecting advection of fresh
urban plumes to the study site. The daily variation of $NO_2$ exhibits a 'U' shape, reaching its minimum value at 13:00, mainly
owing to effects of emission, boundary layer height and photochemical reactions. In the clean period, the daily variation of $PM_{2.5}$
is similar to that of $NO_2$, both showing a 'U' shape, reaching their lowest values at noon. However, during the haze phase, the
daily variation pattern of $PM_{2.5}$ appears somewhat different. There is a noticeable trough in the early morning, remains stable
during the day, and starts to rise after sunset.

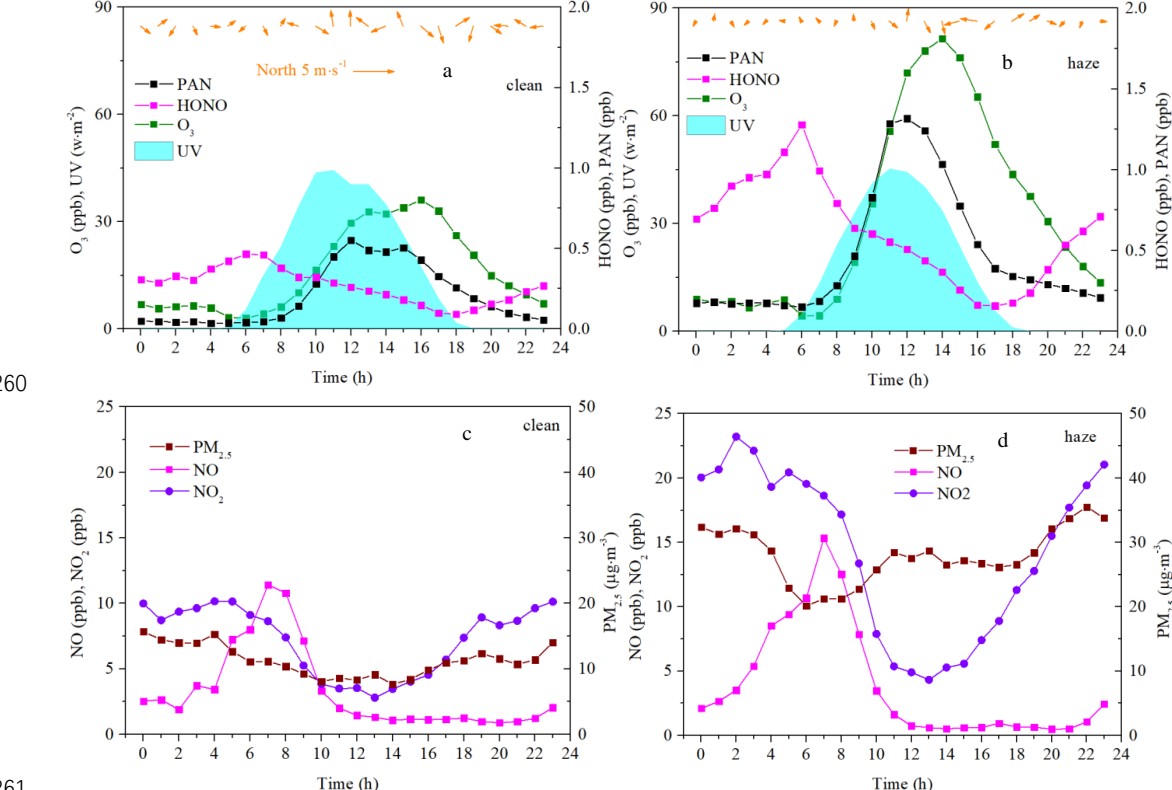



**Figure 2.** The diurnal variations of PAN, HONO, $O_3$, and UV during clean (a) and hazy (b) periods, as well as the diurnal
variations of $PM_{2.5}$, NO, and $NO_2$ during clean (c) and hazy (d) periods.

**3.2 PAN formation: key factors and mechanisms**





To investigate the key factors and mechanisms of PAN formation, PAN was simulated by constraining the MCM-based box
model with meteorological conditions and observed concentrations of precursor gases. The model successfully replicated the
variations in PAN, achieving an IOA of 0.75. (Fig. 3(a)), which was within the accepted range (0.66-0.87) in previous studies
(Zeng et al., 2019). The model captured its formation rate well in general, with observed rates varying from 0.04 to 0.52 ppb·h$^{-1}$
(average: 0.20 ppb·h$^{-1}$) and modeled rates ranging from 0.09 to 0.46 ppb·h$^{-1}$ (average: 0.19 ppb·h$^{-1}$) (see Fig. 3(b)). The similar
result was found in the North China Plain (NCP) region in the wintertime (Xu et al., 2021). When calculating the IOA separately
for clean and hazy periods, it was found that the IOA significantly increased to 0.89 and 0.81 (Fig. 3(c)), respectively. This
phenomenon indicates a substantial difference in the PAN production and destruction mechanisms between clean and hazy
periods. Furthermore, the simulated values are closer to the observed values during clean period, reflected in a higher $R^2$ value
($R^2$=0.6782) and a K value closer to 1 (K=0.9097) (Fig. 3(c)). In contrast, the $R^2$ value and the K value during hazy period are
only 0.4708 and 0.7477, respectively (Fig. 3(c)). This phenomenon suggests that reactions without considered in MCM may
enhance PAN generation during hazy periods.

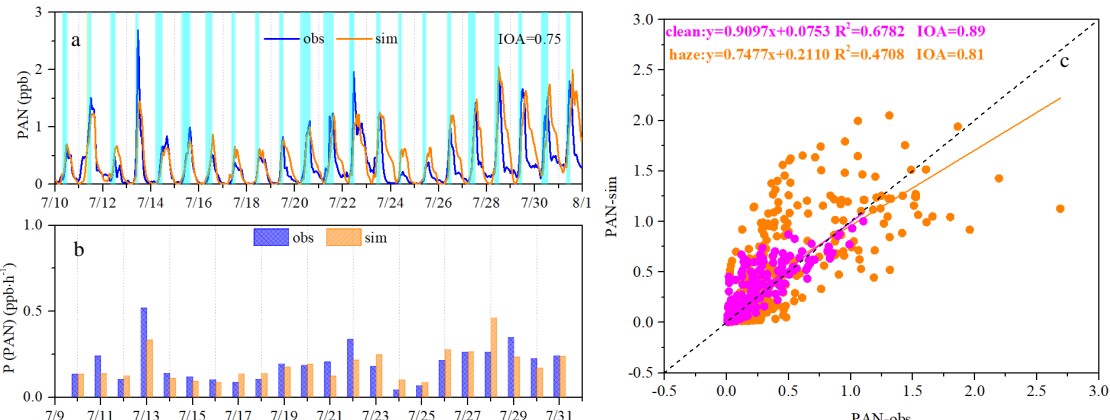


**Figure 3.** Comparisons of modeled PAN and observed PAN (a) variation (daytime photochemical PAN production periods
indicated by cyan shading), (b) production rates, (c) correlation between PAN observations and simulated values

To identify the key factors influencing the performance of the model simulation, we used the difference between the model
simulation values and the observed values (bias) as the target. The remaining variables, which were not input into the OBM
model, such as $NH_3$, $HNO_3$, HCl (alkaline and acidic gaseous pollutants), $PM_{2.5}$ concentrations and their components, as well as
physical process parameters like wind speed and wind direction, were used as features. As shown in Fig. 4 (a), through
XGBoost-SHAP machine learning, we found that $NH_3$ is the most significant parameter affecting bias, contributing 19.68 %. A
scatter plot analysis of the SHAP values of $NH_3$ versus $NH_3$ concentrations revealed that as $NH_3$ concentrations increase (Fig. 4
(b)), the OBM model tends to overestimate more significantly. To date, there are very few studies that directly address the impact
of $NH_3$ on PAN formation. Xu et al. (2021) suggested that $NH_3$ could promote the formation of HONO, which in turn affects



PAN formation. However, since we included HONO as an input to constrain the model, the indirect influence of NH$_3$ on PAN
formation through HONO can be excluded. NH$_3$ in the atmosphere can preferentially react with sulfuric acid (H$_2$SO$_4$) to form
ammonium sulfate ((NH$_4$)$_2$SO$_4$) secondary inorganic aerosols (Behera et al., 2013), leading to the heterogeneous reaction
removal of PAN by secondary inorganic aerosols. This result is validated by the negative correlation between the SHAP values
of NH$_4^+$ and SO$_4^{2-}$ and their respective concentrations. NO$_3^-$ is the second most significant parameter influencing the bias
between the two, contributing 11.33 % (Fig. 4 (a)). NO$_3^-$ has a negative correlation with the bias (Fig. 4 (c)), indicating that
higher NO$_3^-$ levels lead to more significant underestimation by the model, suggesting that NO$_3^-$ promotes PAN formation in the
actual atmosphere (Hanst, 1971). PM$_{2.5}$ is the third most significant parameter (Fig. 4 (a)), contributing 9.4 %. PM$_{2.5}$ has a
positive correlation with the bias (Fig. 4 (d)), indicating that higher PM$_{2.5}$ levels lead to more significant overestimation by the
model, suggesting that PAN can undergo heterogeneous removal on the surface of PM$_{2.5}$ in the actual atmosphere (Sun et al.,

300   2022).

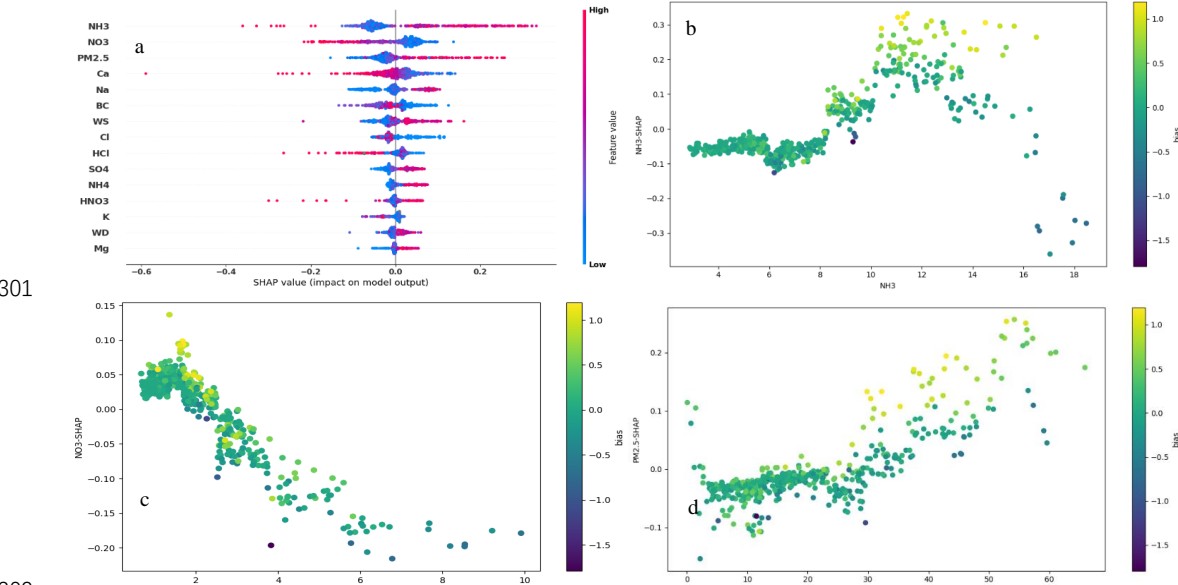

**Figure 4.** Feature importance was obtained by XGBoost-SHAP method (a). The scatter plots between concentration of top three
important features and their SHAP values (b-d), and colored with the difference between the model simulation values and the
observed values (bias).

Figure 5 (a) and (b) show the average production and destruction rates of PAN during clean and haze periods without PAN
constrained. During the haze period, both the production and destruction rates of PAN are significantly higher than during the
clean period. The higher production rate of PAN during the haze period is due to the higher concentration of PAN precursors,
while the higher destruction rate is because both the temperature and PAN concentration are higher. Regarding the net





production rate, it is also higher during the haze period than during the clean period, which corresponds to the previously
observed diurnal variation. From 6:00 to 12:00 during the haze period, the net production rate of PAN is positive, with an
average value of 0.19 ppb·h$^{-1}$. During the clean period, from 6:00 to 12:00, the net production rate of PAN is 0.12 ppb·h$^{-1}$. The
diurnal variation of PAN shows that from 6:00 to 12:00, the average net production rates during the haze and clean periods are
0.20 ppb·h$^{-1}$ and 0.09 ppb·h$^{-1}$, respectively. The model-simulated net production rate is close to the observed net production rate,
further indicating that the model can simulate PAN well, and also confirming that PAN in summer mainly comes from local
production. The net production rate of PAN during the haze period is similar to the summer results in urban areas of the Pearl
River Delta (PRD), which is 0.17 ppb·h$^{-1}$, while the net production rate of PAN during the clean period is similar to the summer
results in rural areas of the PRD, which is 0.12 ppb·h$^{-1}$ (Liu et al., 2024).
Figure 5 (c) and (d) show the average production and destruction rates of PAN during clean and haze periods with PAN
constrained. The net production rate of PAN is approximately zero at night during both clean and haze periods, while there is a
significant difference in the net production rate during the day. During the clean period, the daytime net production rate of PAN is
greater than zero, with an average value of 0.19 ppb·h$^{-1}$. In contrast, during the haze period, the net production rate of PAN is
negative from 6 a.m. to 1 p.m., with an average value of -0.47 ppb·h$^{-1}$, and positive from 2 p.m. to 5 p.m., with an average value
of 0.47 ppb·h$^{-1}$. Previous research has shown that an increase in temperature, an increase in PAN concentration, or a decrease in
PAN precursors (including VOCs and NO$_2$) can cause the net production rate of PAN to change from positive to negative (Liu et
al., 2024). We conducted a correlation analysis of the net production rate of PAN with temperature, PAN concentration, VOCs,
and NO$_2$ and found that the net production rate of PAN had the best correlation with PAN concentration (R$^2$=0.1316), showing a
significant negative correlation (k=-0.5283) (Fig. S7). Additionally, we also observed that when the net production rate of PAN is
negative, the PAN concentration is often very high (Fig. S7). As shown in Fig. 5, we conducted sensitivity experiments by
reducing the PAN concentration by 80 %, i.e., 0.2 times the observed value, and found that the net production rate of PAN was
positive throughout the observation period. Conversely, when the PAN concentration was increased by 140%, i.e., 2.4 times the
observed value, the net production rate of PAN was found to be almost negative throughout the observation period. Besides, we
also conducted sensitivity experiments on temperature and found that when simulating winter temperatures, i.e., 0.4 times the
observed value, with a temperature range of 9.25-15.29°C, the net production rate of PAN was positive throughout the observation
period. Similarly, when simulating spring and autumn temperatures, i.e., 0.6 times the observed value, with a temperature range of
13.87-23.39°C, the net production rate of PAN was also positive throughout the observation period. In conclusion, the net
production rate of PAN becomes negative with PAN constrained, further suggesting the existence of an unknown compensatory
mechanism.



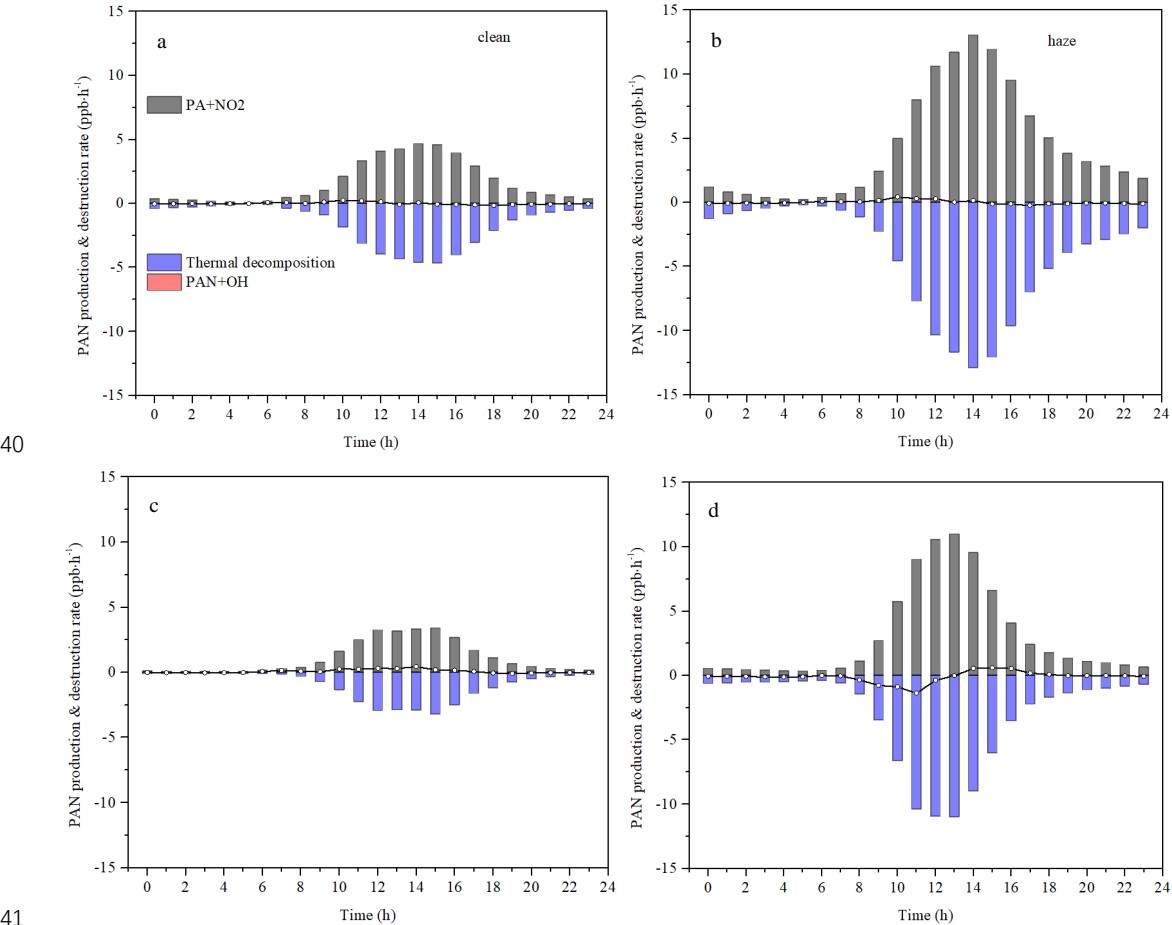

**Figure 5.** Average diurnal variation of the simulated production, destruction and net rates of PAN during clean (a) and haze (b) without PAN constrained. And average diurnal variation of the simulated production, destruction and net rates of PAN during clean (c) and haze (d) with PAN constrained.

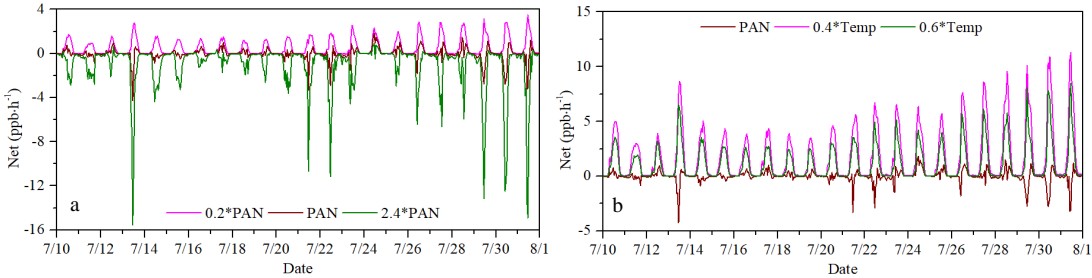

**Figure 6.** Net PAN production rates at different PAN concentrations (a) and different temperatures (b)

PAN is formed when the PA radical reacts with $NO_2$. Given the swift equilibrium between R2 and R4 at high temperatures, budget



analysis of PA's production and consumption pathways is frequently used to detail the mechanisms behind PAN formation (Sun et
al., 2020). Figure 7 illustrates the diurnal patterns of the primary production and loss pathways for the PA radical across different
periods. As shown in Fig. 7, during haze days, the rates of PA production and destruction were twice as high as those on clean
days. This indicates that radical cycling and photochemical formation were more efficient during haze days, driven by higher
temperatures and a greater abundance of precursors (Zeng et al., 2019). The PA radical production rate from PAN thermal
decomposition reached its peak at 15:00 (3.22 ppb·h$^{-1}$) and 13:00 (10.99 ppb·h$^{-1}$) for clean and haze days, perfectly coinciding
with the peak temperature time. In addition, the conversion of PAN into PA radical through thermal decomposition had high
correlations with temperature during both haze ($R^2$=0.82) and clean days ($R^2$=0.77). The conversion of PAN into PA radical
through thermal decomposition during haze days was significantly higher than that during clean days, which was not only
enhanced by higher temperature but also maintained by higher PAN concentration during haze days. The thermal decomposition
of PAN to PA radical during the day (5:00-18:00 local time) accounted for 68.22 % and 45.59 % during haze and clean days,
respectively. The pathways that did not account for the transformation between PA and PAN reached their peak around noon
(11:00 local time), coinciding with the highest solar radiation and the most intense photochemical reactions, which has been
observed in spring and autumn at the same site (Liu et al., 2022).
Production rates of PA from other pathways related to precursors showed single-peak patterns of these four pathways around noon,
which suggested that the PA radical generated from these pathways was primarily increased by intense solar radiation at noontime
(Sun et al., 2020). The average day PA radical production rates from $CH_3CHO$ via reactions with OH and $NO_3$ were 1.10 and 0.93
ppb·h$^{-1}$, accounting for 48.85% and 49.35 % (exclude PAN thermal decomposition sources) during haze and clean days,
respectively. These percentages were comparable to previous studies in Guangzhou (46 %) (Yuan et al., 2018) and Beijing (34.11-
50.19 %) (Xue et al., 2014), suburban site of Chongqing (47.72 %) (Sun et al., 2020). The second production pathway involved
MGLY undergoing photolysis and oxidation through reactions with OH and $NO_3$ (haze: 0.50 ppb·h$^{-1}$ and clean: 0.42 ppb·h$^{-1}$),
contributing to 22.27 % and 22.12 % for haze and clean days, respectively. Subsequently, radical cycling processes—including the
decomposition of RO radicals and the reactions of acyl peroxy radicals with NO—were also significant contributors to PA
production, accounting for 18.98% on haze days and 19.54% on clean days. PA from the other OVOCs (excluding $CH_3CHO$,
MGLY) via photolysis and oxidation reactions involving OH, $NO_3$, and $O_3$, accounted for 9.90 % and 8.99% during haze (0.22
ppb·h$^{-1}$) and clean days (0.17 ppb·h$^{-1}$). There were no notable differences in the proportions of individual pathways contributing to
PA between haze and clean days, indicating comparable pollutant compositions in the atmospheric around IUE (Zeng et al., 2019).
The primary contributor to the PAN destruction rate was the reaction between PA and $NO_2$, accounting for 67.72 % and 51.09 %
during haze (4.74 ppb·h$^{-1}$) and clean days (1.76 ppb·h$^{-1}$), respectively, followed by PA+NO, contributing to 32.28 % and 48.91 %
during haze (2.26 ppb·h$^{-1}$) and clean days (1.69 ppb·h$^{-1}$), respectively.



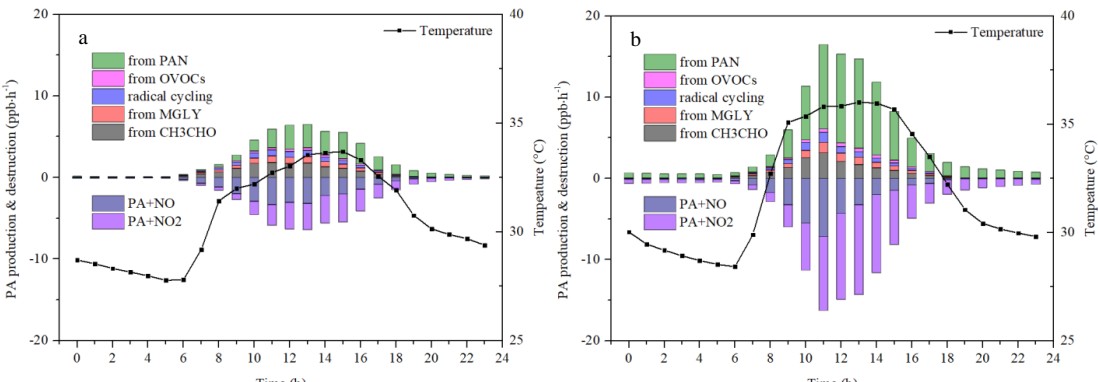

**Figure 7.** PA radical production and destruction pathways on (a) clean days and (b) haze days.

### 3.3 Sensitivity of PAN formation and its impact on the local atmosphere

To determine the principal precursors influencing PAN formation, sensitivity modeling analyses were carried out to investigate how PAN relates to its precursors. The RIR reflects how sensitive PAN formation is to changes in its precursor levels. As shown in Fig.8 (a), decreases in NO led to strong negative RIR ranging from -0.67 to -0.27 (-0.52 ± 0.13) throughout the observation period. However, RIR is positive for other species, with $NO_2$ (0.50 ± 0.11) and VOCs (0.50 ± 0.15) having the highest RIR, followed by HONO (0.12 ± 0.04) and $O_3$ (0.10 ± 0.03). Around 50 types of VOCs were classified as alkanes, OVOCs, halogenated hydrocarbons (Halo), alkenes, aromatics (Arom), and isoprene ($C_5H_8$ representing biogenic hydrocarbons). Among these VOCs, the RIR of alkenes (0.22 ± 0.07) is the highest, followed by $C_5H_8$ (0.13 ± 0.04) and Arom (0.13 ± 0.04), while OVOCs (0.06 ± 0.01) and Halo (0.05 ± 0.01) have very low RIRs (Fig. 8 (b)). These phenomena indicated that increased NO level would inhibit the production of PAN while increased $NO_2$, VOCs (especially alkenes, $C_5H_8$, and Arom), HONO, and $O_3$ would promote the production of PAN. Because the values of NO and $NO_2$ RIR are approximately equal but with opposite signs, the RIR for NOx is almost zero, indicating that the PAN generation at this site is not sensitive to NO*x*. Zeng et al. (2019) also observed that $NO_2$ had a positive effect on PAN formation, while NO had a negative effect, in a suburban area of Hong Kong. This finding aligns with the fact that $NO_2$ directly contributes to PAN production, whereas NO reduces PA radicals, thereby inhibiting PAN formation. Based on the scenario analysis (Empirical Kinetic Modeling Approach (EKMA)), all data points for the 22 days fell above the ridge line (Fig. 8(c)). A reduction in VOCs at these points resulted in lower PAN concentrations, indicating that PAN formation at IUE was influenced by VOCs and thus VOC-sensitive. Our previous research also found that in this coastal city, PAN generation is limited by VOCs during the spring and autumn seasons. The difference is that previous studies indicated that reducing $NO_2$, like reducing NO, also leads to an increase in PAN concentration in spring and autumn (Liu et al., 2022). This is because the NO*x* concentration in spring and autumn is significantly higher than in summer, which is consistent with that both $NO_2$ and NO inhibit the formation of PAN in regions with high NO*x* concentrations (Liu et al., 2024).





We divided the RIRs for different species into haze and clean periods and found that the RIRs during clean periods were

consistently higher than those during haze periods (Fig. 8(d)), which indicated that altering the concentrations of these species

during clean periods had a greater impact on PAN formation. The rapid thermal decomposition of PAN at high temperatures is

likely the primary reason. During the haze period, the main source of PA radical was PAN decomposition, which accounted for

68.22%, and the other sources were smaller than that during the clean period (the source of the PA radical would be demonstrated

in the following paragraph). Therefore, the sensitivity of PAN production to precursors and HONO & $O_3$ producing OH became

lower during the haze period (Liu et al., 2021b; Liu et al., 2022).

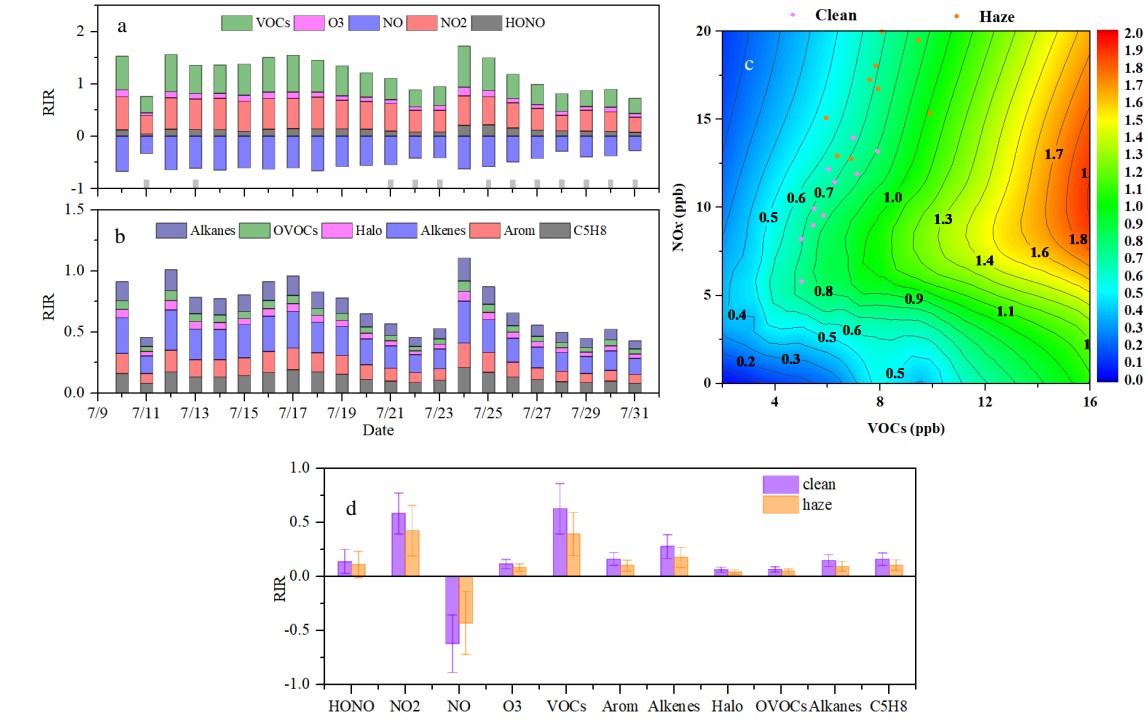

**Figure 8.** These four figures illustrate the RIR of PAN formation to major precursors (a), the impact of different VOCs species (b),

the isopleth diagrams of PAN formation (c), and a comparison of RIRs between clean and polluted periods (d).

As shown in Fig. 9, $\Delta HO_2$ and $\Delta OH$ are positive for most periods, indicating that the PAN mechanism promotes the generation of

$HO_2$ and OH. Over the entire period, $\Delta HO_2$ is $8.43\times10^{-5}$ ppb, with no significant difference between clean and hazy periods, being

$8.18\times10^{-5}$ ppb and $8.64\times10^{-5}$ ppb respectively. OH behaves similarly, with $\Delta OH$ being $4.55\times10^{-7}$ ppb over the entire period, and

also showing no significant difference between clean and hazy periods, being $4.93\times10^{-7}$ ppb and $4.23\times10^{-7}$ ppb respectively. The

increase in simulated OH and $HO_2$ concentrations suggests that PAN photochemistry is enhancing radical formation and AOC at

this site (Liu et al., 2024). Unlike $HO_2$ and OH, $\Delta RO_2$, $\Delta NO_2$, and $\Delta NO$ are negative for most periods, because PAN formation

uses up PA and $NO_2$, the reduction in PA leads to a decrease in the amount of $RO_2$. Over the entire period, $\Delta RO_2$ is $-6.4\times10^{-4}$ ppb,



with no significant difference between clean and hazy periods, being -6.1×10⁻⁴ ppb and -6.5×10⁻⁴ ppb respectively. The average
values of $\Delta NO_2$ and $\Delta NO$ during the entire observation period are -0.17 and -0.01 ppb respectively, with significant differences
between hazy and clean periods. Specifically, $\Delta NO_2$ is -0.22 during hazy periods and only -0.11 during clean periods, indicating
that the PAN mechanism consumes more $NO_2$ during hazy periods. Similarly, $\Delta NO$ is -0.05 during hazy periods, showing an
inhibitory effect, while it is 0.03 during clean periods, showing a promoting effect.

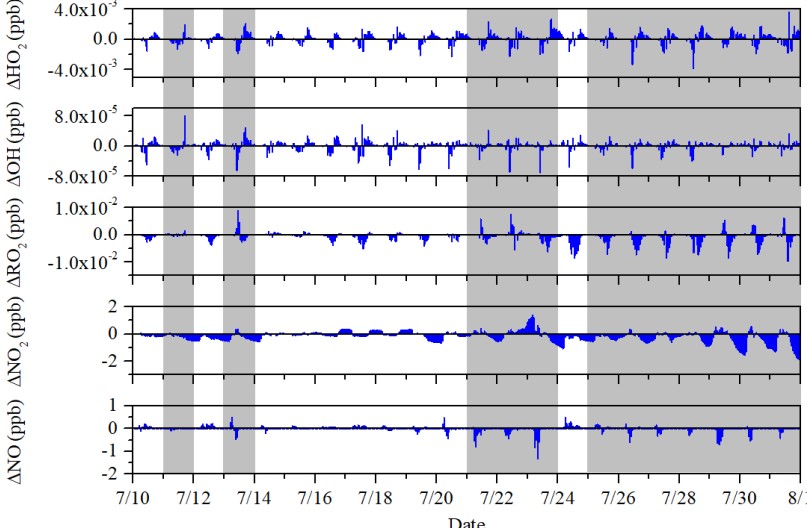


**Figure 9.** The difference of $HO_2$, OH, $RO_2$, $NO_2$, and NO between base scenario with PAN mechanism and scenario without PAN
mechanism

As shown in Fig.10 (a), the PAN mechanism inhibited 85.80% of net ozone production during the entire observation period, with
inhibition rates of 83.75% and 87.50% during clean and haze periods, respectively. This result is consistent with previous spring
observations at the same site, where the inhibition rate was 83% (Liu et al., 2022). The PAN mechanism mainly inhibits the net
ozone generation by increasing the $RO_2+NO_2$ reaction (Fig.10(a)), with negligible impact from other reactions (Fig. S11). As
shown in Fig.10(b), the diurnal variation trend indicates that the PAN mechanism's inhibitory effect on ozone is significantly
greater during haze periods than during clean periods. Additionally, regardless of whether it is during haze periods or clean
periods, the PAN mechanism's inhibitory effect on ozone is significantly greater during the day than at night. These phenomena all
indicate that the higher the PAN concentration, the more pronounced the inhibitory effect of the PAN mechanism on ozone
(Fig.10(c)). Under low precursor conditions, competition among these precursors may limit their secondary formation, thus
resulting in inhibition (Liu et al., 2024).



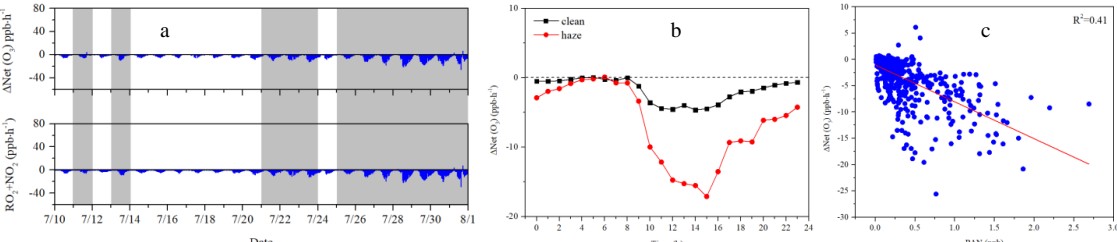

**Figure 10.** (a) Time series plot of $\Delta$Net ($O_3$) and the reaction of $RO_2+NO_2$, (b) Diurnal variation of $\Delta$Net ($O_3$) during clean and

hazy conditions, (c) Correlation between $\Delta$Net ($O_3$) and PAN.

**Conclusion**

This study thoroughly investigated the summertime PAN formation mechanism and established its connection to haze pollution. In

addition to NO and TVOCs, the concentration of all pollutants during the haze period is above twice that during the cleaning

period, indicating that the oxidation of NO and TVOCs during the haze period is stronger, which is conducive to the oxidation of

NO and TVOCs into secondary pollutants, such as $O_3$ and PAN. The slopes of linear regression between the daily maximum

values of PAN and $O_3$ were 0.021 ppb/ppb and 0.009 for clean and hazy condition, respectively, implies that PAN precursors

accounted for only a small fraction of the total VOCs, especially for hazy condition. High temperature should be another factor

contributing to the lower production efficiency of PAN in the southeast coastal region. During the whole observation period, the

IOA=0.75, indicating that the MCM model is well-suited for exploring the photochemical formation of PAN. During the clean

period, simulation results were better than during the haze period ($R^2$: 0.6782 vs. 0.4708, slope K: 0.9097 vs. 0.7477), indicating

that PAN during the haze period may originate from reactions without considered in MCM. Additionally, the net production rate

of PAN becomes negative with PAN constrained. This further indicates that, despite the high temperatures, there is still a

significant concentration of PAN, suggesting the existence of an unknown compensatory mechanism. Through XGBoost-SHAP

machine learning, this unknown compensatory mechanism may involve $NO_3^-$ promoting PAN formation. Both RIR and EKMA

indicate that PAN formation in this region is VOC-controlled. Controlling emissions of VOCs, particularly alkenes, $C_5H_8$, and

aromatics, would be beneficial for mitigating PAN pollution. The RIR results also show that during the clean period, PAN is more

sensitive to changes in various pollutants than during the haze period, highlighting the significant importance of deep emission

reductions. PAN presented the promotion effects on OH and $HO_2$, while inhibited $O_3$ formation, $RO_2$, NO and $NO_2$. This study

improves our thorough understanding of PAN photochemistry and offers valuable scientific guidance for the future management

of PAN pollution.



**Data availability**

The observation data at this site are available from the authors upon request.

**Authorship Contribution Statement**

**Baoye Hu**: Methodology, Formal analysis, Investigation, Data curation, Writing – original draft. **Naihua Chen**: Software, Formal analysis. **Rui Li:** Software, Formal analysis. **Mingqiang Huang**: Software. **Jinsheng Chen**: Funding acquisition, Supervision, Writing - Review & Editing. **Youwei Hong**: Formal analysis. **Lingling Xu**: Investigation. Xiaolong Fan: Investigation. **Mengren Li**: Investigation. **Lei Tong**: Investigation. **Qiuping Zheng**: Investigation. **Yuxiang Yang:** Writing - Review & Editing

**Competing interests**

The authors declare that they have no conflict of interest.

**Acknowledgments**

This work was supported by the National Natural Science Foundation of China (grant nos. 42305102, U22A20578), Natural Science Foundation of Fujian Province (grant nos. 2023J05179), Natural Science Foundation of Zhangzhou City (grant nos. ZZ2023J07), Fujian Provincial Department of Education (grant nos. JAT210279), the Fund of Minnan Normal University President (grant nos. KJ2021009). This study was funded by Xiamen Atmospheric Environment Observation and Research Station of Fujian Province, and Fujian Key Laboratory of Atmospheric Ozone Pollution Prevention (Institute of Urban Environment, Chinese Academy of Sciences).

**Supplementary information**

Attached please find supplementary information associated with this article.

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

Background Site in the Pearl River Delta Region: Production Efficiency and Regional Transport, Aerosol Air Qual. Res., 15,

833-841, 10.4209/aaqr.2014.11.0275, 2015.

Xue, L., Wang, T., Wang, X., Blake, D. R., Gao, J., Nie, W., Gao, R., Gao, X., Xu, Z., Ding, A., Huang, Y., Lee, S., Chen, Y.,

Wang, S., Chai, F., Zhang, Q., and Wang, W.: On the use of an explicit chemical mechanism to dissect peroxy acetyl nitrate

formation, Environ. Pollut., 195, 39-47, 10.1016/j.envpol.2014.08.005, 2014.

Xue, L. K., Wang, T., Guo, H., Blake, D. R., Tang, J., and Zhang, X. C.: Sources and photochemistry of volatile organic

compounds in the remote atmosphere of western China: results from the Mt. Waliguan Observator, Atmos. Chem. Phys., 13,

8551-8567, 10.5194/acp-13-8551-2013, 2013.

Yang, X., Wu, K., Wang, H., Liu, Y., Gu, S., Lu, Y., Zhang, X., Hu, Y., Ou, Y., Wang, S., and Wang, Z.: Summertime ozone

pollution in Sichuan Basin, China: Meteorological conditions, sources and process analysis, Atmos. Environ., 226,

10.1016/j.atmosenv.2020.117392, 2020.

Ye, C., Zhang, N., Gao, H., and Zhou, X.: Photolysis of Particulate Nitrate as a Source of HONO and NO$x$, Environ. Sci. Technol.,

51, 6849-6856, 10.1021/acs.est.7b00387, 2017.

Yuan, J., Ling, Z., Wang, Z., Lu, X., Fan, S., He, Z., Guo, H., Wang, X., and Wang, N.: PAN–Precursor Relationship and Process

Analysis of PAN Variations in the Pearl River Delta Region, Atmos., 9, 10.3390/atmos9100372, 2018.

Yukihiro, M., Hiramatsu, T., Bouteau, F., Kadono, T., and Kawano, T.: Peroxyacetyl nitrate-induced oxidative and calcium

signaling events leading to cell death in ozone-sensitive tobacco cell-line, Plant Signal Behav., 7, 113-120,

10.4161/psb.7.1.18376, 2012.

Zeng, L., Fan, G. J., Lyu, X., Guo, H., Wang, J. L., and Yao, D.: Atmospheric fate of peroxyacetyl nitrate in suburban Hong Kong

and its impact on local ozone pollution, Environ. Pollut., 252, 1910-1919, 10.1016/j.envpol.2019.06.004, 2019.

Zhai, S., Jacob, D. J., Franco, B., Clarisse, L., Coheur, P., Shah, V., Bates, K. H., Lin, H., Dang, R., Sulprizio, M. P., Huey, L. G.,

Moore, F. L., Jaffe, D. A., and Liao, H.: Transpacific Transport of Asian Peroxyacetyl Nitrate (PAN) Observed from Satellite:

Implications for Ozone, Environ. Sci. Tech., 58, 9760-9769, 10.1021/acs.est.4c01980, 2024.



Zhang, G., Mu, Y., Zhou, L., Zhang, C., Zhang, Y., Liu, J., Fang, S., and Yao, B.: Summertime distributions of peroxyacetyl nitrate

(PAN) and peroxypropionyl nitrate (PPN) in Beijing: Understanding the sources and major sink of PAN, Atmos. Environ.,

103, 289−296, 10.1016/j.atmosenv.2014.12.035, 2015.

Zhang, J., Guo, Y., Qu, Y., Chen, Y., Yu, R., Xue, C., Yang, R., Zhang, Q., Liu, X., Mu, Y., Wang, J., Ye, C., Zhao, H., Sun, Q.,

Wang, Z., and An, J.: Effect of potential HONO sources on peroxyacetyl nitrate (PAN) formation in eastern China in winter, J.

Environ. Sci. (China), 94, 81-87, 10.1016/j.jes.2020.03.039, 2020.

Zhang, J. M., Wang, T., Ding, A. J., Zhou, X. H., Xue, L. K., Poon, C. N., Wu, W. S., Gao, J., Zuo, H. C., Chen, J. M., Zhang, X.

C., and Fan, S. J.: Continuous measurement of peroxyacetyl nitrate (PAN) in suburban and remote areas of western China,

Atmos. Environ., 43, 228-237, 10.1016/j.atmosenv.2008.09.070, 2009.

Zhu, J., Wang, S., Wang, H., Jing, S., Lou, S., Saiz-Lopez, A., and Zhou, B.: Observationally constrained modeling of atmospheric

oxidation capacity and photochemical reactivity in Shanghai, China, Atmos. Chem. Phys., 20, 1217-1232, 10.5194/acp-20-

1217-2020, 2020.

