# Peer review of "Understanding summertime peroxyacetyl nitrate (PAN) formation and"

_EGUsphere, 2024_

## Referee Comment (RC3)

Hu et al. made PAN observations and investigated its summertime formation with the aid of box modelling and machine learning. The paper provides valuable insights into the summertime formation of PAN and its link to aerosol pollution, which has been an unsolved issue during recent years. The following issues should be addressed before it can be considered for publication.

Major issues:

1. L76-81: The authors should probably further emphasize that Xiamen is a coastal site and give a background understanding on pollution as well as climate characteristics in Xiamen. What differs Xiamen from the sites where PAN was previously already investigated? This might help emphasizing the importance of this study.

2. Sect. 2.1 VOCs measurements were not introduced in terms of instrumentation details and observed species. Figures present TVOCs concentrations, how was VOCs constrained within the MCM model if you did not have the individual VOCs species.

3. L190-192: Does it make sense to correlate daily maximum BC and PAN, when obviously they peaked at very different times of day? BC usually peaks during nighttime under low boundary layer conditions, while PAN peaks during noontime before $O_3$ due to strong thermal deposition losses. If there were any correlation between BC and PAN, you should at least prove it with a correlation analysis that uses data from the same time of day.

4. L213-214: In addition to O3 and PAN formation, a great part of TVOCs might have turned into SOA.

5. L231: It would be better if you added the standard deviations to the averaged values.

6. L239-245: If you corrected for thermal losses, would this change the slope of PAN vs. O3 production?

7. L258-259, Fig.2: The wind direction varied differently during clean and haze periods, are daytime northerly winds connected to pollution transport? There was a rise in PM2.5 during prenoon hours during haze days, was that connected to stronger secondary formation or transport processes?

8. L287-291: NH3 and HONO often reveal very high correlations in urban regions due to the influence of common vehicle emissions. Was that also the case for Xiamen? Since both were considered in the model, the model must have selected only one variable, would results be different if only HONO and no NH3 were included? The uptake of aqueous uptake of PAN was introduced to be very weak, what mechanisms do you believe led to strong uptake of PAN on ammoniumnitrate aersols?

9. L294-297: How does NO3- promote PAN formation? Might it be common enhanced formation of NO3- and PAN during atmospheric processes that led to these results?

10. L321-327: If I am understanding things correctly, constraining PAN within the model would lead to the following results: if constraints are larger than model estimates, the model would add to thermal degradation losses leading to lower net

production and vice versa. Since the model performed fairly well in simulating PAN production and could relatively accurately reflect its atmospheric level, why were there negative net production during haze conditions, when PAN was constrained? Since temperature and precursor constraints were the same, do you suggest that constrained concentrations were higher than those simulated by the model? However, simulated PAN was often higher than observed ones when there were no constraints. Isn't that in contradiction? Adjusting PAN constraints to 0.2 times that of actual values is far below those modelled without PAN constraints, why?

11. L379: I recommend a brief summary on which factors played the dominant role in boosting PA production rates on haze days.

Minor issues:

1. L22-24: Grammatically incorrect, please rephrase.
2. L27-28: The number of valid digits should be unified across the manuscript.
3. L76: "Ximen"➔"Xiamen"
4. L85-86: Grammatically incorrect, please rephrase.
5. L161-162: Grammatically incorrect, please rephrase.
6. L177: by "daily maximum average" do you mean "maximum daily average"?
7. L218: Mt. Waliguan is a global background station.
8. Fig 8c: It is quite difficult to differentiate between clean and haze dots without enlarging the figure, please select colors with larger contrasts.
9. Figure labels are often too small and hard to read.

---

## Author Comment (AC1)

**General Overview:**

The manuscript (egusphere-2024-2631) presents an attractive aspect of the connection of peroxyacetyl nitrate (PAN) to summertime haze and photochemical air pollution. As claimed by the manuscript, summertime PAN formation is quite an important topic. The topic also falls within the scope of the journal Atmospheric Chemistry and Physics (ACP). The authors present data from field observations with high temporal resolution as well as those from a box model and a machine learning model. The authors not only discuss the key factors and mechanisms of PAN formation based on the presented data but also tested the sensitivity of PAN to several other chemical species in the atmosphere. This manuscript is laid out well and shows the knowledge gap it fills. This manuscript is recommended to be published after addressing the concerns and comments below with minor revisions.

Response: Thank you for your thorough review and constructive comments on our manuscript (egusphere-2024-2631). We are delighted to hear that you find our study on the connection of peroxyacetyl nitrate (PAN) to summertime haze and photochemical air pollution to be significant, and that it falls within the scope of the journal Atmospheric Chemistry and Physics (ACP). We also appreciate your recognition of our data analysis and model applications. In response to your feedback, we have carefully considered and made revisions as follow. In general replies, we use blue font; red font indicates parts added in the revised manuscript, and blue italic font denotes references.

**Major Concerns:**

Line 227 – 229: It is stated in the manuscript that "The daily variation of PAN exhibits a clear unimodal pattern, with concentrations starting to rise after sunrise and decreasing after 12:00 caused by thermal decomposition of PAN at high temperatures". However, the diurnal variations of PAN, $O_3$, and UV during the clean period as shown in Figure 2 (a) seems to exhibit bimodal patterns, which might not be consistent with and might not support the statement in the manuscript.

Response: Thank you for your valuable comment. We believe that the bimodal patterns of PAN and $O_3$ during the clean period are primarily influenced by UV radiation. As shown in Figure 2(a), UV also exhibits a bimodal pattern during this period, which likely contributes to the similar behavior of PAN and $O_3$ concentrations. In contrast, during haze conditions, UV levels remain relatively stable, resulting in the absence of bimodal patterns for PAN and $O_3$. We have clarified this point in the revised manuscript to ensure consistency and accuracy in our discussion: Although PAN and $O_3$ exhibit a slight bimodal pattern during the clean period, this is primarily due to the bimodal pattern of UV during this time.

Line 276 – 277: It is stated in the manuscript that "reactions without considered in MCM may enhance PAN generation during hazy periods". This statement seems to be based on the fact that the slope for the hazy period in Figure 3 (c) is less than 1. However, this statement might be challenged by the facts that the $R^2$ value is only 0.4708 and that there are multiple simulated PAN concentrations higher than observed PAN concentrations, which make the statement less convincing. It would probably be safer and more convincing to state that some reactions related to PAN generation might be missing in the MCM during the hazy period.

Response: Thank you for your insightful suggestion. You are correct that our initial statement was derived from the slope being less than 1. However, since the intercept is a positive value (0.21) and some simulated PAN concentrations are indeed greater than the observed values, we acknowledge that it would be more accurate to state that "some reactions related to PAN generation or destruction might be missing in the MCM during the hazy period." We have revised the manuscript to reflect this clarification.

Line 385 – 387 and 391 – 393: It is stated in Line 385 – 387 that "decreases in NO led to strong negative RIR" and in Line 391 – 393 that "increased NO level would inhibit the production of PAN". These 2 statements seem to be contradictory to each other. It would be great if these 2 statements could be explained in more detail to make sure that the statements in this manuscript are consistent with each other.

Response: Thank you for pointing out this potential inconsistency. According to the RIR calculation formula (eq. S4 in the supporting information), a negative RIR indicates that after a 20% reduction in NO, the net production rate of PAN is higher than that before the reduction. This means that reducing NO can indeed promote PAN generation, making the two statements not contradictory. To avoid any ambiguity, we have revised Lines 385–387 to state: the RIR of NO was negative, ranging from -0.67 to -0.27 (-0.52 ± 0.13) throughout the observation period.

Line 457 – 458: The expression of the sentence "This further indicates that, despite the high temperatures, there is still a significant concentration of PAN, suggesting the existence of an unknown compensatory mechanism" seems unclear when it is put right after the sentence "Additionally, the net production rate of PAN becomes negative with PAN constrained". It might take the reader quite some time and efforts to see if these 2 sentences are logically connected. It might be worth trying to rephrase the sentences to make the expressions more clearly.

Response: Thank you for your constructive feedback. We understand that the connection between these two sentences may not be immediately clear. Because of the high temperatures in summer, if PAN concentrations remain high, this would lead to a large removal rate of PAN by thermal decomposition, ultimately resulting in a negative net production rate of PAN. To enhance readability, we have rephrased this sentence: However, the observed increasing PAN concentrations indicates that the actual net production rate is positive, suggesting that there are additional sources contributing to PAN generation that are not considered in the MCM mechanism.

**Minor Concerns:**

Line 31 – 32: Under the context of the abstract, the meanings of acronyms RIR and EKMA are not clear. The full form "Relative Incremental Reactivity" of RIR is not shown until Section 2.2 (Line 129) while the full form "Empirical Kinetic Modeling Approach" of EKMA is not shown until Section 3.3 (Line 397). It would be great if the full form can appear first in the manuscript before a corresponding acronym is used.

Response: Thank you for your feedback. The full forms of "relative incremental reactivity (RIR)" and "empirical kinetic modeling approach (EKMA)" have been provided the first time they are mentioned in the manuscript. We appreciate your attention to this matter.

Line 33 and 462: The terminology "deep emission reduction" appears in the manuscript without clear definition. If the definition is similar to that in some previous studies, it would be great to cite the relevant studies with clear definition. Otherwise, it might be helpful to define it in the manuscript.

Response: The term "deep emission reduction" is a technical term that refers to significant efforts in emissions reduction. This term is widely applied in the context of carbon emissions (Deetman et al., 2014), and it can also be used for the reduction of other pollutants.

References

Deetman, S., Hof, A. F., & van Vuuren, D. P. (2014). Deep $CO_2$ emission reductions in a global bottom-up model approach. Climate Policy, 15(2), 253–271. https://doi.org/10.1080/14693062.2014.912980

Line 43: It might be better to cite the source of reactions R1 – R3 when they first appear in the description.

Response: Thank you for your suggestion. We have cited the source of reactions R1–R3 at their first appearance in the manuscript to provide proper attribution and enhance clarity for the readers.

Line 58 – 59: While discussing previous studies on wintertime photochemical air pollution in the manuscript, it would be helpful to cite the source of the statement "it is found that aerosol promotes PAN generation".

Response: Thank you for your suggestion. We have added the appropriate citation for the statement "it is found that aerosol promotes PAN generation" in the revised manuscript. This would provide proper context and support for the discussion on previous studies regarding photochemical air pollution in wintertime.

Line 59 – 60: While discussing previous studies on wintertime photochemical air pollution in the manuscript, please "Surprisingly high concentrations of OH radical, particularly under hazy conditions, have been observed and are largely attributed to HONO photolysis".

Response: Added.

Line 139, 283, and 288: The meaning of the acronym OBM is not clear. The full name of the acronym seems to be missing in the manuscript.

Response: Thank you for pointing this out. We have included the full name of the acronym "OBM" in the revised manuscript at the first instance it appears.

Line 144: It might be better if the source of the "SHapley Additive exPlanations (SHAP)" approach can be cited.
Response: Thank you for your suggestion. We have added a citation for the "Shaply Additive explanation (SHAP)" approach in the revised manuscript.

Line 144 – 146: It might be better if some of the studies that have successfully applied the SHAP approach can be cited.
Response: Thank you for your recommendation. We have included citations for several studies that had successfully applied the SHAP approach in the revised manuscript.

Line 188 – 189: The statement "the precursor concentration of PAN is significantly lower than in the northern region" is not quite clear. Is it meant to be It would be helpful if the statement can be clarified.
Response: Thank you for your feedback. We have clarified the statement to specify that "the precursor concentrations of PAN, including $NO_2$ and VOCs, are significantly lower in the studied area compared to those in the northern region."

Line 190 – 191figure to support the statement "The correlation between the daily maximum values of PAN and BC is the strongest (R=0.85), followed by $O_3$ (R=0.75)"?
Response: Thank you for your suggestion. We have included Fig. R1 in the revised manuscript to visually support the statement "The correlation between the maximum daily values of PAN and BC is the strongest (R=0.85), followed by $O_3$ (R=0.75)".

[Figure]

Fig. R1 The correlation between the maximum daily values of PAN and BC (a), as well as the correlation between the maximum daily values of PAN and $O_3$ (b).

Line 227: It would be helpful if it could be pointed out that the average diurnal patterns of PAN and related variables for clean and hazy conditions are shown in Figure 2 before the contents in Figure 2 are discussed in detail without clearly stating where the contents are shown.
Response: Thank you for your suggestion. We have added 'Fig. 2' after the sentence 'The average diurnal patterns of PAN and related variables have been averaged separately for clean and hazy conditions'.

Line 274 – 276: Both the $R^2$ and K values are discussed in the manuscript, but only the $R^2$ values are defined (Line 152) and shown (Figure 3 (c)). The K values seem to be not defined in the manuscript. It might take the reader some time and efforts to notice that the K values potentially mean the slopes in Figure 3 (c). I would suggest the authors to clearly define
Response: Thank you for your suggestion. The K has been defined in the revised manuscript: Furthermore, the simulated values are closer to the observed values during clean period, reflected in a higher $R^2$ value ($R^2$=0.6782) and a slope value (K) closer to 1 (K=0.9097) (Fig. 3(c)).

Line 279 – 280 (Figure 3): The legend of the figure shows "obs" and "sim" without their definitions. if the legend of the figure

could be defined in and be consistent with the caption of the figure. For example, the caption could be modified as "Comparison of observed (obs) PAN and simulated (sim) PAN".

Response: Thank you for your suggestion. We agree that clarifying the definitions of "obs" and "sim" in the figure legend would improve consistency and understanding. We have modified the caption to read: Comparison of observed (obs) PAN and simulated (sim) PAN, ensuring that the definitions are clearly stated.

Line 282 – 283 and 304 – 305: It might be better if the definition of bias described as "difference between the model simulation values and the observed values" can be expressed mathematically as what values minus what values for the reader to be clear about the mathematic definition.

Response: Thank you for your suggestion. We appreciate the need for clarity, and we have specified that the bias is calculated as the model simulation minus the observed value.

Line 283 and 285: Since there are 2 models, a box model and a machine learning model, being used in this study

Response: Thank you for your suggestion. We aim to use a machine learning model to evaluate the reasons behind the biases in the box model simulations. To avoid ambiguity, we have added 'OBM' in the sentence: To identify the key factors influencing the performance of the OBM model simulation.

Line 285 – 286: It is stated that "$NH_3$ is the most significant parameter affecting bias, contributing 19.68 %". However, it seems that the number 19.68 % is not shown in Figure 4 (a). It would be helpful to clarify whether the contribution of 19.68 % is on average or obtained in some other ways. It would also be great to show such a value in the corresponding figure as described in the manuscript.

Response: Thank you for your suggestion. This proportion is calculated by taking the absolute values of the SHAP values for all features, summing them up, and then dividing the absolute SHAP value of a particular feature by the total sum. In other words, it represents the average proportion of the absolute SHAP value for each feature during the whole observation period.

[Figure]

Fig. R2 The average proportion of the absolute SHAP value for each feature during the whole observation period.

Line 294 – 295: It is stated that "$NO_3^-$ is the second most significant parameter influencing the bias between the two, contributing 11.33 %". However, it seems that the number 11.33 % is not shown in Figure 4 (a). Could the authors clarify whether the contribution of 11.33 % is on average or obtained in some other ways? it be possible to show such a value in the corresponding figure as described in the manuscript?

Response: Same as above.

Line 297: It is stated that "$PM_{2.5}$ is the third most significant parameter, contributing 9.4 %". However, it seems that the number 9.4 % is not shown in Figure 4 (a). It would be helpful to clarify whether the contribution of 9.4 % is on average or

obtained in some other ways. It would also be great to show such a value in the corresponding figure as described in the manuscript.

Response: Same as above.

Line 307 – 308 and 320 – 321: Since there are data from field observations and model simulations being used in this study, it would be helpful if it could be clearly stated whether the average production and destruction rates of PAN during clean and haze periods are observed or simulated by what model. Although the caption of Figure 5 mentions that they are simulated, it would be great if it could be clearly stated in the paragraph of description as well.

Response: Thank you for your helpful comment. We would like to clarify that the production and destruction rates of PAN mentioned in Lines 307 – 308 are results from the OBM model simulations without the constraint of observed PAN values. In contrast, the rates discussed in Lines 320 – 321 are also from the OBM model simulations but include the constraint of observed PAN values. We have revised the manuscript to clearly state this distinction in the caption of Figure 5: Average diurnal variation of the OBM simulated production, destruction and net rates of PAN during clean (a) and haze (b) without PAN constrained. And average diurnal variation of the OBM simulated production, destruction and net rates of PAN during clean (c) and haze (d) with PAN constrained. Additionally, we have reiterated this information in the relevant paragraph to ensure consistency and clarity: Figure 5 (a) and (b) show the average production and destruction rates of PAN during clean and haze periods, as simulated by OBM without PAN constrained. Figure 5 (c) and (d) show the average production and destruction rates of PAN during clean and haze periods, as simulated by OBM with PAN constrained.

Line 312 – 313: I would appreciate it whether the net production rate of PAN is simulated net production rate or observed net production rate.

Response: Thank you for your question. The net production rate of PAN mentioned in Lines 312 – 313 is based on the model simulation results. We have ensured this is clearly stated in the revised manuscript: From 6:00 to 12:00 during the haze period, the simulated net production rate of PAN is positive, with an average value of 0.19 ppb·h$^{-1}$. During the clean period, from 6:00 to 12:00, the simulated net production rate of PAN is 0.12 ppb·h$^{-1}$.

Line 313 – 314: Is the diurnal variation of PAN based on observation or simulations? I would suggest the authors to it here.

Response: The result in Lines 313 – 314 is based on observations. We have included this clarification in the revised manuscript: The observed diurnal variation of PAN shows that from 6:00 to 12:00, the average net production rates during the haze and clean periods are 0.20 ppb·h$^{-1}$ (Fig. 2(a)) and 0.09 ppb·h$^{-1}$ (Fig. 2(b)), respectively.

Line 327 – 329: It is stated that "We conducted a correlation analysis of the net production rate of PAN with temperature, PAN concentration, VOCs, and NO$_2$". However, it seems that only the correlation between the simulated net production rate of PAN and observed PAN concentration is shown in Figure S7. It would be great if the other correlations mentioned in the manuscript can be provided to support the statement.

Response: Thank you for your valuable feedback. We acknowledge that only the correlation between the simulated net production rate of PAN and observed PAN concentration was shown in Figure S7. To address this, we have included additional correlation analyses for the net production rate of PAN with temperature, VOCs, and NO$_2$ in the revised manuscript as Fig. R3. These results would provide further support for our claims.

[Figure]

[Figure]

[Figure]

Fig. R3 Correlation analysis of the net production rate of PAN with temperature (a), PAN (b), TVOCs (c), and $NO_2$ (d) concentration, respectively

Line 330 – 332: It is stated that the sensitivity experiments are shown in Figure 5, but it seems that the sensitivity experiments are actually shown in Figure 6.

Response: Thank you for pointing this out. We have clarified that the sensitivity experiments are indeed shown in Figure 6, not Figure 5, and we have corrected this in the revised manuscript.

Line 331, 333, 335, 337, and 347 (Figure 6): Could the authors state in the text and caption of the figure whether the net production rate of PAN is simulated net production rate of PAN or not?

Response: Thank you for your suggestion. We have clarified in the text and the figure caption that the net production rate of PAN refers to the simulated net production rate.

Line 349 – 350: It is stated that "budget analysis of PA's production and consumption pathways is frequently used". However, only 1 study is cited to support this statement, which might not be convincing. It might be better if more studies are cited to support the statement that the method is frequently used. Otherwise, it might be safer and more convincing to state that the method has been used with only 1 citation.

Response: Thank you for your valuable feedback. I acknowledge that citing only one study may not convincingly support the statement regarding the frequency of budget analysis for PA's production and consumption pathways. I have revised the manuscript to include additional studies that demonstrate the widespread use of this method: Given the swift equilibrium between R2 and R4 at high temperatures, budget analysis of PA's production and consumption pathways are frequently used to detail the mechanisms behind PAN formation (Sun et al., 2020; Liu et al., 2022a; Liu et al., 2024).

Line 351: It would be helpful if it is clearly stated whether the diurnal patterns are simulated diurnal patterns or not.

Response: Thank you for your insightful comment. We have clarified in the revised manuscript that the diurnal patterns presented are simulated patterns: Figure 7 illustrates the diurnal patterns of the primary production and loss pathways for the PA radical simulated by OBM across different periods.

Line 356 – 357: It is stated that "the conversion of PAN into PA radical through thermal decomposition had high correlations with temperature during both haze and clean days". It would be appreciated if a figure of correlations can be provided to support the statement.

Response: Thank you for your suggestion. We have included Fig. R4 showing the correlation between PAN thermal decomposition and temperature. From the figure, it can be observed that during clean days, the temperature is linearly correlated with PAN thermal decomposition, with an $R^2$ value of 0.82; however, when using an exponential correlation, the $R^2$ increases to 0.95. Similarly, during haze days, the linear correlation has an $R^2$ of 0.77, but the exponential correlation significantly improves it to 0.91. Therefore, in the revised manuscript, we have changed the statement to: the conversion of PAN into PA radical through thermal decomposition had high exponential correlations with temperature during both haze ($R^2$=0.91) and clean days ($R^2$=0.95) (Fig. S13).

[Figure]

Fig. R4 Correlation between temperature and PAN thermal decomposition during clean (a) and haze (b) period.

Line 364 – 366: It would be clearer to the reader if it could be stated what "these four pathways" mentioned in the statement are. The pathways are shown in Figure 7, but it would be clearer if they could be described in the paragraph as well.

Response: Thank you for your valuable feedback. We appreciate your suggestion to clarify what "these four pathways" are. We have included a brief description of each pathway in the paragraph to enhance clarity for the reader: Production rates of PA from other pathways related to precursors, including OVOCs, radical cycling, MGLY, and $CH_3CHO$, showed single-peak patterns around noon, which suggested that the PA radical generated from these pathways was primarily increased by intense solar radiation at noontime (Sun et al., 2020).

Line 377: It is stated that "The primary contributor to the PAN destruction rate was the reaction between PA and $NO_2$", but it seems that this sentence is meant to describe the PA destruction rate instead of the PAN destruction rate.

Response: Thank you for pointing this out. We have revised the sentence to clarify that it refers to the PA destruction rate rather than the PAN destruction rate.

Line 381 (Figure 7): Are the PA radical production and destruction rates simulated or not?

Response: Thank you for your question. Yes, both the PA radical production and destruction rates were simulated in our study. We have clarified this point in the revised manuscript to avoid any confusion: Figure 7. PA radical production and destruction pathways simulated by OBM on (a) clean days and (b) haze days.

Line 416 and 421: It is stated that "$\Delta HO_2$ and $\Delta OH$ are positive for most periods" and that "$\Delta RO_2$, $\Delta NO_2$, and $\Delta NO$ are negative for most periods". It might be better if there are specific numbers to quantitatively support the statements since there are also multiple periods with negative values of $\Delta HO_2$ and $\Delta OH$ and multiple periods with positive values of $\Delta RO_2$, $\Delta NO_2$, and $\Delta NO$ shown in Figure 9.

Response: Thank you for your suggestion. We have included specific numerical values to quantitatively support the statements regarding $\Delta HO_2$, $\Delta OH$, $\Delta RO_2$, $\Delta NO_2$, and $\Delta NO$. This would provide clearer support for the trends described and address the variability shown in Figure 9: As shown in Fig. 9, $\Delta HO_2$ and $\Delta OH$ are positive for most periods, accounting for 72.16% and 70.83%, respectively, indicating that the PAN mechanism promotes the generation of $HO_2$ and OH. Over the entire period, $\Delta HO_2$ is $8.43 \times 10^{-5}$ ppb (Table S1), with no significant difference between clean and hazy periods, being $8.18 \times 10^{-5}$ ppb and $8.64 \times 10^{-5}$ ppb respectively. OH behaves similarly, with $\Delta OH$ being $4.55 \times 10^{-7}$ ppb over the entire period, and also showing no significant difference between clean and hazy periods, being $4.93 \times 10^{-7}$ ppb and $4.23 \times 10^{-7}$ ppb respectively (Table S1). The increase in simulated OH and $HO_2$ concentrations suggests that PAN photochemistry is in favor of radical formation and AOC at this site (Liu et al., 2024). Unlike $HO_2$ and OH, $\Delta RO_2$ and $\Delta NO_2$ are negative for most periods, accounting for 53.22% and 67.23%, respectively, because PAN formation uses up PA and $NO_2$, the reduction in PA leads to a decrease in the amount of $RO_2$. Over the entire period, $\Delta RO_2$ is $-6.45 \times 10^{-4}$ ppb, with no significant difference between clean and hazy periods, being $-6.11 \times 10^{-4}$ ppb and $-6.5 \times 10^{-4}$ ppb respectively (Table S1). The average value of $\Delta NO_2$ during the entire observation period is -0.17 ppb respectively, with significant differences between hazy and clean periods (Table S1). Specifically, $\Delta NO_2$ is -0.22 during hazy periods and only -0.11 during clean periods, indicating that the PAN mechanism consumes more $NO_2$ during hazy periods. Although $\Delta NO$ is positive for most periods, accounting for 78.79%, the overall mean is -0.01, with significant differences between hazy and clean periods (Table S1). $\Delta NO$ is -0.05 during hazy periods, showing an inhibitory effect, while

it is 0.03 during clean periods, showing a promoting effect.

Line 417, 419, 423, and 424: The statistic term "significant difference" appears multiple times in the manuscript. It would be more convincing if the significance levels to determine whether there would be statistically significant differences or not are clearly stated in this study.

Response: Thank you for your comment. I have included the results of the independent samples T-test in the revised manuscript, as shown in Table R1. From the table, it is evident that $\Delta NO_2$ and $\Delta NO$ exhibit significant differences between the clean and hazy periods at the 0.01 significance level, while $HO_2$, $OH$, and $RO_2$ do not show significant differences.

Table R1. The independent samples T-test between haze and clean period

|  | Haze (mean±stdev) | Clean (mean±stdev) |
|---|---|---|
| $\Delta HO_2$ | $8.64 \times 10^{-5} \pm 8.49 \times 10^{-4}$ | $8.18 \times 10^{-5} \pm 5.76 \times 10^{-4}$ |
| $\Delta OH$ | $4.23 \times 10^{-7} \pm 1.37 \times 10^{-5}$ | $4.94 \times 10^{-7} \pm 1.49 \times 10^{-5}$ |
| $\Delta RO_2$ | $-6.55 \times 10^{-4} \pm 2.28 \times 10^{-3}$ | $-6.11 \times 10^{-4} \pm 1.43 \times 10^{-3}$ |
| $\Delta NO_2$ | $-0.22 \pm 0.48^{**}$ | $-0.11 \pm 0.27$ |
| $\Delta NO$ | $-0.05 \pm 0.17^{**}$ | $0.03 \pm 0.09$ |

Note: ** The significance level is 0.01 between haze and clean period.

Line 420: The meaning of the acronym AOC is not clear. The full name of the acronym seems to be missing in the manuscript.

Response: Thank you for your comment. The full name of the acronym AOC, "atmospheric oxidative capacity," have been added to the manuscript to ensure clarity.

Line 429 – 430: the definition of "The difference of $HO_2$, $OH$, $RO_2$, $NO_2$, and $NO$ between base scenario with PAN mechanism and scenario without PAN mechanism" could be expressed mathematically as what values minus what values for the reader to be clear about the mathematic definition.

Response: Thank you for your comment. We have clarified the mathematical definition by explicitly stating that it represents the values from the base scenario with the PAN mechanism minus the values from the scenario without the PAN mechanism: **Figure 9.** The time series of $\Delta HO_2$, $\Delta OH$, $\Delta RO_2$, $\Delta NO_2$, and $\Delta NO$. The $\Delta HO_2$, $\Delta OH$, $\Delta RO_2$, $\Delta NO_2$, and $\Delta NO$ is calculated as the base scenario with the PAN mechanism minus the scenario without the PAN mechanism.

Line 432 – 434: The term "inhibition rate" appears in the manuscript without clear definition. It would be helpful to define it in the manuscript.

Response: Thank you for your comment. We have added a clear definition of "inhibition rate" in the manuscript to ensure clarity: As shown in Fig.10 (a), the PAN mechanism inhibited 85.80% of net ozone production during the entire observation period, with inhibition rates (the percentage of negative $\Delta Net (O_3)$) of 83.75% and 87.50% during clean and haze periods, respectively. This result is consistent with previous spring observations at the same site, where the inhibition rate was 83% (Liu et al., 2022). $\Delta Net (O_3)$ is calculated as the base scenario with the PAN mechanism minus the scenario without the PAN mechanism.

Line 440: It would be clearer if it could be stated that the precursors mentioned here are precursors of what specific chemical species.

Response: Thank you for your suggestion. We have clarified that the precursors mentioned in line 440 specifically refer to $NOx$ and VOCs: Under the condition of low precursors (including $NOx$ and VOCs), competition among these precursors may limit their secondary transformation, thus resulting in inhibition (Liu et al., 2024).

Line 440: It would be clearer if it could be stated what "their" means in the sentence with the term "their secondary formation".

Response: Thank you for your feedback. "Their" refers to these precursors. To avoid ambiguity, I have revised this sentence to:Under the condition of low precursors (including $NOx$ and VOCs), competition among these precursors may limit their secondary transformation, thus resulting in inhibition (Liu et al., 2024).

Line 451: It would be clearer if the units for the number 0.009 could be stated.

Response: Thank you for your comment. The unit for the number 0.009 is also ppb/ppb, and we have added this unit to the revised manuscript for clarity.

Line 456: It would be helpful if it is clearly stated whether the net production rate of PAN is simulated net production rate of PAN or not.

Response: Thank you for your suggestion. We have clarified that the net production rate of PAN mentioned is the simulated net production rate of PAN. This has been explicitly stated in the revised manuscript.

Figure S6 term "their product" in the caption should be clearly described along with the mathematic expression $O_3 \times JO_1D$ to be consistent with the axis labels of the figure.

Response: Revised.

Figure S7: It seems that the axis labels of abscissa and ordinate are missing.

Response: Added and shown as Fig. R3.

Figure S8: It might be clearer for the reader to see the variations of time series if the axis limits could be adjusted closer to the minimum and maximum of each time series.

Response: Thank you for your feedback. The current axis limits were set to emphasize that these reactions are almost negligible compared to the $RO_2 + NO_2$ reaction. As shown in Fig. R5, even when we minimize the range of the vertical axis as much as possible and apply the same range to other sources, we still find that the variations from other sources are nearly negligible (Fig. R6).

[Figure]

Fig. R5 Time series plot of $\Delta$Net ($O_3$) and the reaction of $\Delta(RO_2+NO_2)$. These values are calculated as the base scenario with the PAN mechanism minus the scenario without the PAN mechanism.

[Figure]

Fig. R6 Time series plot of the reaction of Δ(HO$_2$+NO), Δ(RO$_2$+NO), Δ(O$_3$/NO$_3$+VOCs), Δ(O$_3$ photolysis), Δ(O$_3$+OH), Δ(O$_3$+HO$_2$), and Δ(OH+NO$_2$). These values are calculated as the base scenario with the PAN mechanism minus the scenario without the PAN mechanism.

**Technical Comments:**

Line 76: "Ximen" seems to be a typographical error of "Xiamen".

Response: Revised.

Line 199 – 201: The sentence "PM$_{2.5}$ concentrations during the haze period were significantly higher than during the clean period, being 2.49 times that of the clean period" might be corrected as "PM$_{2.5}$ concentrations during the haze period were significantly higher than those during the clean period, being 2.49 times those of the clean period".

Response: Revised.

Line 204 – 205: The sentence "During the haze period, ozone concentrations were also significantly higher than during the clean period, being 2.04 times that of the clean period" might be corrected as "During the haze period, ozone concentrations were also significantly higher than those during the clean period, being 2.04 times those of the clean period".

Response: Revised.

Line 435: The "Fig. S11" seems to be meant as "Fig. S8" since there are only 8 figures in the supporting information.

Response: Revised.

Figure S5: The axis label of the abscissa "maximum daily ozone concentration (PAN)" seems to be a typographical error of "maximum daily ozone concentration (O$_3$)".

Response: Revised as Fig. R7

[Figure]

Fig. R7 Correlation between PAN and O$_3$ maximum daily concentrations during haze and clean.

---

## Author Comment (AC2)

• Line 59 – 60: While discussing previous studies on wintertime photochemical air pollution in the manuscript, please cite the source for the statement "Surprisingly high concentrations of OH radical, particularly under hazy conditions, have been observed and are largely attributed to HONO photolysis".

Response: Added.

• Line 188 – 189: The statement "the precursor concentration of PAN is significantly lower than in the northern region" is not quite clear. Is it meant to be "the precursor concentration of PAN is significantly lower than that in the northern region" or "the precursor concentration of PAN is significantly lower in the northern region"? It would be helpful if the statement can be clarified.

Response: Thank you for your feedback. We have clarified the statement to specify that "the precursor concentrations of PAN, including $NO_2$ and VOCs, are significantly lower in the studied area compared to those in the northern region."

• Line 190 – 191: Could the authors provide a figure to support the statement "The correlation between the daily maximum values of PAN and BC is the strongest (R=0.85), followed by O3 (R=0.75)"?

Response: Thank you for your suggestion. We have included Fig. R1 in the revised manuscript to visually support the statement "The correlation between the maximum daily values of PAN and BC is the strongest (R=0.85), followed by $O_3$ (R=0.75)".

• Line 274 – 276: Both the R2 and K values are discussed in the manuscript, but only the R2 values are defined (Line 152) and shown (Figure 3 (c)). The K values seem to be not defined in the manuscript. It might take the reader some time and efforts to notice that the K values potentially mean the slopes in Figure 3 (c). I would suggest the authors to clearly define the K values in the manuscript.

Response: Thank you for your suggestion. The K has been defined in the revised manuscript: Furthermore, the simulated values are closer to the observed values during clean period, reflected in a higher $R^2$ value ($R^2$=0.6782) and a slope value (K) closer to 1 (K=0.9097) (Fig. 3(c)).

• Line 279 – 280 (Figure 3): The legend of the figure shows "obs" and "sim" without their definitions. It would be great if the legend of the figure could be defined in and be consistent with the caption of the figure. For example, the caption could be modified as "Comparison of observed (obs) PAN and simulated (sim) PAN".

Response: Thank you for your suggestion. We agree that clarifying the definitions of "obs" and "sim" in the figure legend would improve consistency and understanding. We have modified the caption to read: Comparison of observed (obs) PAN and simulated (sim) PAN, ensuring that the definitions are clearly stated.

• Line 283 and 285: Since there are 2 models, a box model and a machine learning model, being used in this study, it would be appreciated if the "target" and the "features" mentioned here are of which model can be clearly stated.

Response: Thank you for your suggestion. We aim to use a machine learning model to evaluate the reasons behind the biases in the box model simulations. To avoid ambiguity, we have added 'OBM' in the sentence: To identify the key factors influencing the performance of the OBM model simulation.

• Line 294 – 295: It is stated that "NO3- is the second most significant parameter influencing the bias between the two, contributing 11.33 %". However, it seems that the number 11.33 % is not shown in Figure 4 (a). Could the authors clarify whether the contribution of 11.33 % is on average or obtained in some other ways? Would it be possible to show such a value in the corresponding figure as described in the manuscript?

Response: Thank you for your suggestion. This proportion is calculated by taking the absolute values of the SHAP values for all features, summing them up, and then dividing the absolute SHAP value of a particular feature by the total sum. In other words, it represents the average proportion of the absolute SHAP value for each feature during the whole observation period.

[Figure]

Fig. R1 The average proportion of the absolute SHAP value for each feature during the whole observation period.

- Line 312 – 313: I would appreciate it if it could be clearly stated whether the net production rate of PAN is simulated net production rate or observed net production rate.

Response: Thank you for your question. The net production rate of PAN mentioned in Lines 312 – 313 is based on the model simulation results. We have ensured this is clearly stated in the revised manuscript: From 6:00 to 12:00 during the haze period, the simulated net production rate of PAN is positive, with an average value of 0.19 ppb·h$^{-1}$. During the clean period, from 6:00 to 12:00, the simulated net production rate of PAN is 0.12 ppb·h$^{-1}$.

- Line 313 – 314: Is the diurnal variation of PAN based on observation or simulations? I would suggest the authors to clearly state it here.

Response: The result in Lines 313 – 314 is based on observations. We have included this clarification in the revised manuscript: The observed diurnal variation of PAN shows that from 6:00 to 12:00, the average net production rates during the haze and clean periods are 0.20 ppb·h$^{-1}$ (Fig. 2(a)) and 0.09 ppb·h$^{-1}$ (Fig. 2(b)), respectively.

- Line 331, 333, 335, 337, and 347 (Figure 6): Could the authors clearly state in the text and caption of the figure whether the net production rate of PAN is simulated net production rate of PAN or not?

Response: Thank you for your suggestion. We have clarified in the text and the figure caption that the net production rate of PAN refers to the simulated net production rate.

- Line 381 (Figure 7): Are the PA radical production and destruction rates simulated or not?

Response: Thank you for your question. Yes, both the PA radical production and destruction rates were simulated in our study. We have clarified this point in the revised manuscript to avoid any confusion: Figure 7. PA radical production and destruction pathways simulated by OBM on (a) clean days and (b) haze days.

- Line 429 – 430: It would be helpful if the definition of "The difference of HO2, OH, RO2, NO2, and NO between base scenario with PAN mechanism and scenario without PAN mechanism" could be expressed mathematically as what values minus what values for the reader to be clear about the mathematic definition.

Response: Thank you for your comment. We have clarified the mathematical definition by explicitly stating that it represents the values from the base scenario with the PAN mechanism minus the values from the scenario without the PAN mechanism: **Figure 9.** The time series of ΔHO₂, ΔOH, ΔRO₂, ΔNO₂, and ΔNO. The ΔHO₂, ΔOH, ΔRO₂, ΔNO₂, and ΔNO is calculated as the base scenario with the PAN mechanism minus the scenario without the PAN mechanism.

- Figure S6: The term "their product" in the caption should be clearly described along with the mathematic expression $O_3 \times JO_1D$ to be consistent with the axis labels of the figure.

Response: Revised.

---

## Author Comment (AC3)

Hu et al. made PAN observations and investigated its summertime formation with the aid of box modelling and machine learning. The paper provides valuable insights into the summertime formation of PAN and its link to aerosol pollution, which has been an unsolved issue during recent years. The following issues should be addressed before it can be considered for publication.

Response: Thank you for your feedback. We appreciate your acknowledgment of the insights provided in our paper regarding the summertime formation of PAN and its connection to aerosol pollution. We have carefully addressed the issues you raised to ensure the manuscript quality for publication standards. In general replies, we use blue font; red font indicates parts added in the revised manuscript, and blue italic font denotes references.

Major issues:

1. L76-81: The authors should probably further emphasize that Xiamen is a coastal site and give a background understanding on pollution as well as climate characteristics in Xiamen. What differs Xiamen from the sites where PAN was previously already investigated? This might help emphasizing the importance of this study.

Response: Thank you for your feedback. We would like to emphasize that Xiamen presents several unique characteristics that distinguish it from other sites where PAN has been previously studied. Firstly, Xiamen is one of the fastest urbanizing regions in southeast China while also being recognized as one of the cities with the best air quality in China. Its air quality could be seen as a model for the future of other urban regions in China. This makes Xiamen a particularly interesting site for studying PAN, as it offers insight into atmospheric chemistry in a rapidly urbanizing yet relatively clean environment. Geographically, Xiamen is located in a low-latitude coastal area, receiving abundant sunlight and long daylight hours during the summer. This results in strong solar radiation and rapid photochemical conversion rates, which differ significantly from the conditions at many inland or higher-latitude PAN study sites. Furthermore, the city is situated in the East Asian monsoon region, acting as a transport channel for atmospheric pollutants from the Yangtze River Delta and Pearl River Delta regions. This means that Xiamen experiences pollution that is overlapped by both local generation and regional transport. Thirdly, Xiamen′s summer climate is influenced by complex meteorological conditions, including typhoons and the West Pacific Subtropical High (WPSH). The WPSH, in particular, creates conditions conducive to the formation and accumulation of photochemical pollutants and particulate matter (Wu et al., 2019). This contrasts with previous PAN studies conducted in regions with less dynamic weather systems. The combination of high temperatures, high humidity, and intense solar radiation, especially in July, likely accelerates both the formation and consumption rates of PAN, making Xiamen an ideal natural laboratory for studying PAN dynamics under high ozone conditions. These factors highlight the importance of this study in understanding the formation mechanisms of PAN in coastal, rapidly urbanizing regions with diverse meteorological influences. We have made the corresponding additions in the revised manuscript as follows (in red font): Xiamen is one of the fastest urbanizing regions in the southeast China and is also one of the coastal cities with the best air quality in China, where the air quality could represent the future of other Chinese urban regions. Between 2018 and 2023, Xiamen ranked among the top 10 cities in China, achieving positions of 7th in 2018, 4th in both 2019 and 2020, 6th in 2021, 9th in 2022, and returning to 7th in 2023 (mee.gov.cn, last assessed October 30, 2014). Xiamen is located in a low-latitude coastal area, with abundant sunlight and long daylight hours during the summer, resulting in strong solar radiation and rapid photochemical conversion rates. The city is typically influenced by the East Asian monsoon and serves as a transport channel for atmospheric pollutants from both the Yangtze River Delta and Pearl River Delta regions. Additionally, during the summer, Xiamen is often affected by complex meteorological conditions such as typhoons and the West Pacific Subtropical High (WPSH). The WPSH creates weather conditions that promote the formation and accumulation of photochemical pollutants and particulate matter (Wu et al., 2019). This setting provides an ideal "laboratory" for investigating the complexities of summertime PAN formation and its relationship with aerosol pollution under high ozone concentrations. In summer, especially in July, high temperatures, high humidity, and intense radiation are likely to accelerate both the formation and consumption rates of PAN.

2. Sect. 2.1 VOCs measurements were not introduced in terms of instrumentation details and observed species. Figures present TVOCs concentrations, how was VOCs constrained within the MCM model if you did not have the individual VOCs species.

Response: Thank you for pointing this out. The VOC measurements were indeed conducted using a gas chromatography mass spectrometer (GC-FID/MS, TH-300B, Wuhan, China) at an hourly time resolution. This included key VOC species such as

alkanes, alkenes, aromatics, and oxygenated VOCs. Detailed information regarding the VOC detection system and calibration procedures is available in our previous study (Liu et al., 2022). Regarding the constraint of VOCs within the MCM model, while the figures in the manuscript present the total VOCs (TVOCs) concentrations, the MCM simulations were constrained by the measured concentrations of individual VOC species from the GC-FID/MS dataset. The individual VOC species were used to initialize and constrain the model inputs. This approach ensures that the MCM model represents the real atmospheric chemistry accurately, based on the observed VOC species. We have included this information in the revised manuscript to clarify the VOC measurements as follow: The VOC measurements were conducted using a gas chromatography mass spectrometer (GC-FID/MS, TH-300B, Wuhan, China) at an hourly time resolution. Detailed information regarding the VOC detection system and calibration procedures is available in our previous study (Liu et al., 2022b).

*Reference:*

*Liu, T., Hong, Y., Li, M., Xu, L., Chen, J., Bian, Y., Yang, C., Dan, Y., Zhang, Y., Xue, L., Zhao, M., Huang, Z., and Wang, H.: Atmospheric oxidation capacity and ozone pollution mechanism in a coastal city of southeastern China: analysis of a typical photochemical episode by an observation-based model, Atmos. Chem. Phys., 22, 2173-2190, 10.5194/acp-22-2173-2022, 2022.*

3.    L190-192: Does it make sense to correlate daily maximum BC and PAN, when obviously they peaked at very different times of day? BC usually peaks during nighttime under low boundary layer conditions, while PAN peaks during noontime before $O_3$ due to strong thermal deposition losses. If there were any correlation between BC and PAN, you should at least prove it with a correlation analysis that uses data from the same time of day.

Response: Thank you for this insightful comment. We fully agree that BC and PAN typically peak at different times of the day, with BC usually reaching its maximum during nighttime under low boundary layer conditions, and PAN peaking around noon due to strong photochemical activity. This temporal mismatch indeed leads to poor direct correlation (with a correlation coefficient of only 0.097) when using hourly data. By using the daily maximum, these short-term fluctuations can be smoothed out, showing more clearly the correlations between pollutants and trends in atmospheric chemical processes.

4.    L213-214: In addition to $O_3$ and PAN formation, a great part of TVOCs might have turned into SOA.

Response: We recognize that a significant portion of TVOCs can lead to secondary organic aerosol (SOA) formation. However, our study focuses on $O_3$ and PAN formation, and since SOA data was lack, we did not include it in our analysis. To avoid ambiguity, we have added the following statement in the revised manuscript: Although it is acknowledged that VOCs can also be converted into SOA, the discussion of SOA is beyond the scope of this study.

5.    L231: It would be better if you added the standard deviations to the averaged values.

Response: We appreciate your suggestion and have added the standard deviations to the averaged values in the revised manuscript. The standard deviations for the haze and clean periods are 0.44 and 0.21, respectively.

6.    L239-245: If you corrected for thermal losses, would this change the slope of PAN vs. $O_3$ production?

Response: Considering thermal losses alters the slope of PAN versus $O_3$ production. Specifically, during haze conditions, the slope increased from 0.009 to 0.1581, and in cleaner periods, it rose from 0.021 to 0.1504 (Fig. R1). This substantial change indicates that the low PAN generation efficiency in our region is largely influenced by thermal decomposition.

[Figure]

Fig. R1. Correlation between PAN and O₃ maximum daily concentrations during haze and clean (a), and correlation between PAN+TPAN and O₃ maximum daily concentrations during haze and clean (b)

7.  L258-259, Fig.2: The wind direction varied differently during clean and haze periods, are daytime northerly winds connected to pollution transport? There was a rise in PM$_{2.5}$ during prenoon hours during haze days, was that connected to stronger secondary formation or transport processes?

Response: The wind direction indeed varied differently during clean and haze periods (Fig. R2). During clean periods, the wind was primarily from the northeast (Fig. R2(a)), while during the haze period, the wind direction was dominated by other directions except for the northeast, particularly the southeast wind (Fig. R2(b)). When the northeast wind blew with high speed, the PM$_{2.5}$ concentration was often low, indicating that the northeast wind primarily acted to clear pollutants, rather than transporting pollution from other regions to the area (Fig. R2(c)). Sulfur oxidation rate and nitrogen oxidation rate (defined as SOR=SO$_4^{2-}$/(SO$_4^{2-}$+SO$_2$) and NOR=NO$_3^-$/(NO$_3^-$+NO$_2$)) were commonly used to represent secondary formation of PM$_{2.5}$. During the haze period around noon, the NOR is significantly greater than that during the clean period (Fig. R3(a)), while the SOR is notably higher after 11 o'clock during the haze period compared to the clean period (Fig. R3(b)). However, the wind speed is always greater during the clean period than during the haze period (Fig. R3(c)). The haze is relatively in a stable state, so the reason of the increase in PM$_{2.5}$ during the haze period is more likely to secondary transformation rather than the transport process. If the transport process were dominant, then the PM$_{2.5}$ concentration would be higher during the clean period when the wind speed is greater.

[Figure]

Fig. R2. Polar frequency of wind speed and wind direction for clean (a) and haze period (b), polar plot of PM$_{2.5}$ concentrations for whole observation period (c)

[Figure]

Fig. R3. The diurnal variation of NOR(a), SOR(b), and WS(c)

8. L287-291: $NH_3$ and HONO often reveal very high correlations in urban regions due to the influence of common vehicle emissions. Was that also the case for Xiamen? Since both were considered in the model, the model must have selected only one variable, would results be different if only HONO and no $NH_3$ were included? The uptake of aqueous uptake of PAN was introduced to be very weak, what mechanisms do you believe led to strong uptake of PAN on ammonium nitrate aerosols?

Response: We conducted a correlation analysis between $NH_3$ and HONO, and found that when using all data, the $R^2$ between the two is only 0.13. However, when focusing only on the morning rush hours (6, 7, and 8 AM), the $R^2$ increases to 0.60 (Fig. R4). This phenomenon suggests that in Xiamen, HONO and $NH_3$ are likely both emitted from motor vehicles. The concentration of $NH_3$ in urban environments has significantly increased due to the over-reduction of NO$x$ in catalytic converters used in automobile exhaust systems (Behera et al., 2013). The OBM model only considered HONO as a constraint, as $NH_3$ is not included in the default MCM mechanism. To investigate whether $NH_3$ affects PAN formation, we used machine learning to incorporate variables such as $NH_3$, $PM_{2.5}$, ws (wind speed), and wd (wind direction), which were not input into the OBM model. The difference between the OBM model simulation values and the observed values was used as the target for exploration. The previous study (Pratap et al., 2021) demonstrated that when the pH of the solution is 4.3, ammonium sulfate solution can promote approximately 30% more gaseous organic compounds to dissolve in the solution compared to pure water. Additionally, with constant pH, the higher the concentration of ammonium sulfate solution, the stronger the promotion effect has. It's a pity that this literature does not discuss the specific mechanism; future research could focus on this mechanism.

[Figure]

Fig. R4 The scatter plot of HONO and $NH_3$, where black hollow circles represent all hourly data, and red solid circles represent morning peak hours (6, 7, and 8 AM).

9. L294-297: How does $NO_3^-$ promote PAN formation? Might it be common enhanced formation of $NO_3$- and PAN during atmospheric processes that led to these results?

Response: A PAN-forming rection involving $NO_3$ is the following:

$$CH_3COO + NO_3 \rightarrow PAN$$

The above $NO_3$ reaction predicts a direct proportion between the concentration of $NO_3$ and the rate of PAN formation, which is a relationship demonstrated experimentally as plotted in Fig. R5(a). However, I previously misunderstood; the $NO_3$ here refers to a radical, not nitrate.

Yes, it is indeed possible that a common source or physical processes contribute to the enhanced formation of both $NO_3^-$ and

PAN during atmospheric processes. Nitrate is predominantly formed from the gaseous precursor NO$x$ through secondary reactions in the atmosphere, which is similar to the main formation pathway of PAN, as PAN is produced during the oxidation of volatile organic compounds in the presence of NOx. We conducted a correlation analysis between PAN and NO$_3^-$ and found a significant positive correlation at the 0.01 level, with a correlation coefficient of 0.374. Additionally, both PAN and NO$_3^-$ exhibit daily variations, peaking around noon (Fig. R5(b)), suggesting they may share a common source. We have made corresponding modifications in the revised manuscript: considering the significant positive correlation between PAN and NO$_3$ at the 0.01 level, with a correlation coefficient of 0.374, and the fact that both reach their peaks around noon (Fig. S11), it is likely that they have a common source. Thank you for your question and for suggesting this possibility, which makes our assumptions and conclusions more convincing.

[Figure]

Fig. R5 First order dependence of PAN formation on NO$_3$ concentration (a) (this figure is from the literature Hanst 1971), diurnal variation of PAN and NO$_3^-$ (b).

10. L321-327: If I am understanding things correctly, constraining PAN within the model would lead to the following results: if constraints are larger than model estimates, the model would add to thermal degradation losses leading to lower net production and vice versa. Since the model performed fairly well in simulating PAN production and could relatively accurately reflect its atmospheric level, why were there negative net production during haze conditions, when PAN was constrained? Since temperature and precursor constraints were the same, do you suggest that constrained concentrations were higher than those simulated by the model? However, simulated PAN was often higher than observed ones when there were no constraints. Isn't that in contradiction? Adjusting PAN constraints to 0.2 times that of actual values is far below those modelled without PAN constraints, why?

Response: Your understanding is correct. Constraining PAN with observational values would lead to similar results: if the observational values are higher than the model estimates, the model would account for the increased losses by thermal degradation, resulting in lower net production; conversely, if the observational values are lower than the estimates, the model would reduce losses by thermal degradation, leading to higher net production. This aligns with the PAN formation and loss processes. The model generally performed well in simulating PAN production and reflecting its atmospheric levels, achieving an IOA of 0.75. However, the simulation performance was lower during haze periods compared to clean periods. During clean periods, the linear fit between the observed and simulated values had an R$^2$ of 0.68, with a K value of 0.91, while during haze periods, the R$^2$ dropped to 0.47, and the K value decreased to 0.75. From Fig. R6(a), we can see that the net PAN production rates with PAN constrained turn negative when PAN concentrations are high, and most of these occurrences are during haze periods. To provide a clearer view of how the net PAN production rate is influenced by the constrained PAN concentrations, we created the scatter plot in Fig. R6(b). From Fig. R6(b), we can see that the larger the observed PAN values exceed the simulated values, the more the net production rate with PAN constraints falls below the rate without PAN constraints. The case without PAN constraints can essentially be viewed as constrained by the simulated values. When the observed PAN values are lower than the simulated values, the net production rate constrained by the observed PAN is higher than that constrained by the simulated values. Additionally, the net PAN production rate constrained by 0.2 times the observed values is significantly higher than the net production rate without PAN constraints (Fig. R7). This is also because the lower the PAN

concentration, the smaller the thermal degradation, which leads to a higher net PAN production rate.

[Figure]

Fig. R6 Time series of observed PAN, simulated PAN, net PAN production rate with PAN constraints, and net PAN production rate without PAN constraints (a), with the shaded areas indicating haze periods; scatter plot of the difference between observed PAN and simulated PAN (ΔPAN) versus the difference between net PAN production rates with and without PAN constraints (ΔNet) (b).

[Figure]

Fig. R7 Net PAN production rates simulated by OBM with 0.2 times the observed PAN values as constraints and without PAN constraints.

11. L379: I recommend a brief summary on which factors played the dominant role in boosting PA production rates on haze days.

Response: Thank you for your valuable feedback. The thermal decomposition of PAN played the dominant role in boosting PA production rates on haze days, followed by from $CH_3CHO$, MGLY, radical cycling and other OVOCs. And we have added the following sentence in the revised manuscript: In summary, the thermal decomposition of PAN played the dominant role in boosting PA production rates during both clean and haze periods, followed by contributions from $CH_3CHO$, MGLY, radical cycling, and other OVOCs.

Minor issues:

1. L22-24: Grammatically incorrect, please rephrase.

Response: Thank you for pointing this out. I have rephrased the section to correct the grammatical errors. The revised text now reads: Notably, PAN has been observed at unexpectedly high concentrations (maximum: 3.04 ppb) during the summertime. The daily maximum values of PAN showed a stronger correlation with black carbon (BC) (R=0.85) than with ozone ($O_3$) (R=0.75), suggesting a close connection between summertime haze and photochemical pollution.

2. L27-28: The number of valid digits should be unified across the manuscript.

Response: Thank you for your valuable feedback. We have retained two decimal places for the data in lines 27 and 28, and changed the four decimal places in other sections of the manuscript to two decimal places.

3. L76: "Ximen"→"Xiamen"

Response: Revised.

4. L85-86: Grammatically incorrect, please rephrase.

Response: Thank you for pointing this out. I have rephrased the section to correct the grammatical errors. The revised text now reads: Using machine learning with XGBoost, we identified the key factors that affect the OBM model's simulation results and clarified the mechanisms linking haze pollution to photochemical air pollution, as indicated by PAN and $O_3$.

5. L161-162: Grammatically incorrect, please rephrase.

Response: Thank you for pointing this out. I have rephrased the section to correct the grammatical errors. The revised text now reads: Combined with the synoptic situation shown in Fig. S4, the 8th typhoon of 2018, Typhoon Maria, made landfall on the morning of the 11th at Huangqi Peninsula in Lianjiang County, Fujian.

6. L177: by "daily maximum average" do you mean "maximum daily average"?

Response: Yes, it refers to the maximum daily average. We have made the corresponding correction in the revised manuscript.

7. L218: Mt. Waliguan is a global background station.

Response: Thank you for raising this point. Mt. Waliguan is indeed a Global Atmosphere Watch (GAW) background station. I have made the corresponding revision in the manuscript accordingly.

8. Fig 8c: It is quite difficult to differentiate between clean and haze dots without enlarging the figure, please select colors with larger contrasts.

Response: Thank you for the suggestion. I have updated the figure with colors that have larger contrasts and increased the size of the dots to better differentiate between clean and haze points (Fig. R8).

[Figure]

Fig. R8 The isopleth diagrams of PAN formation

9. Figure labels are often too small and hard to read.

Response: Thank you for your observation. We have increased the size of the figure labels to improve readability in the revised manuscript.

*Reference*

*Behera, S. N., Sharma, M., Aneja, V. P., and Balasubramanian, R.: Ammonia in the atmosphere: a review on emission sources, atmospheric chemistry and deposition on terrestrial bodies, Environ. Science and Pollution Research, 20, 8092-8131, 10.1007/s11356-013-2051-9, 2013.*

*Liu, T., Chen, G., Chen, J., Xu, L., Li, M., Hong, Y., Chen, Y., Ji, X., Yang, C., Chen, Y., Huang, W., Huang, Q., and Wang, H.: Seasonal characteristics of atmospheric peroxyacetyl nitrate (PAN) in a coastal city of Southeast China: Explanatory factors and photochemical effects, Atmos. Chem. Phys., 22, 4339-4353, 10.5194/acp-22-4339-2022, 2022.*

*Pratap, V., Carlton, A. G., Christiansen, A. E., and Hennigan, C. J.: Partitioning of Ambient Organic Gases to Inorganic Salt Solutions: Influence of Salt Identity, Ionic Strength, and pH, Geophysical Research Letters, 48, 10.1029/2021gl095247, 2021.*

---

## Author Comment (AC4)

**General Overview:**

This manuscript reports the deep connection between the Peroxyacetyl nitrate (PAN) formation with summertime haze and photochemical air pollution in Xiamen. The authors have used multiple observations of trace gases, aerosol chemical composition, and meteorological parameters, combined with a model simulation with the Master Chemical Mechanism (MCMv3.3.1) and a machine learning model. The manuscript is well written and I recommend its publication after addressing the major and minor comments given below in order for it to be published.

Response: Thank you for your thorough review and positive feedback on our manuscript. We appreciate your insightful comments and have addressed both the major and minor points you raised to enhance the clarity and quality of our work. In general replies, we use blue font; red font indicates parts added in the revised manuscript, and blue italic font denotes references.

**Major comments**

Line 198. The main text describes two periods: "haze" and "clean" conditions based on the maximum diurnal hourly concentration ($> 35\ \mu g/m^3$). I believe another condition should be considered, as the period from July 26 to 31 shows distinct characteristics compared to other periods (e.g., 11 and 13 of July). This behavior remains consistent over time, indicating increased pollution and greater atmospheric reactivity (more precursors, radicals, etc.), which is reflected in the concentrations of $O_3$, $NO_2$, and $PM_{2.5}$. I think this period should be treated as a special event, as something significant occurred during that time.

Response: Yes, the "haze" and "clean" conditions are based on the maximum daily concentration ($> 35\ \mu g \cdot m^{-3}$). The period from July 26 to 31 was continuously influenced by the West Pacific Subtropical High (WPSH), providing relatively stable (ws = 1.04 $m \cdot s^{-1}$), high-temperature (maximum daily average of 37.82 °C), high-humidity (maximum daily average of 81.65 %), and strong UV(maximum daily average of 46.71$W \cdot m^{-2}$) atmospheric conditions, which are conducive to the accumulation and secondary transformation of pollutants. As a result, we observed an increase in the concentrations of various pollutants (including $O_3$, $NO_2$, and $PM_{2.5}$) during this period. On July 13, the Xiamen also affected by WPSH. The days of July 11 and 21-23 were influenced by the periphery of typhoons. This indicates that the rise in $PM_{2.5}$ concentrations was not solely due to the WPSH. Although July 12-15 was also influenced by the WPSH, there was no significant increase in pollutant concentrations, likely because the impact of Typhoon No. 9—"Shanshen" began on the 16th. If we categorize based solely on the WPSH, it would diverge from the theme of this paper.

One factor that could be related to the "clean" and "haze" conditions is wind direction, which differs between the period of July 14 to 19, considered as "clean conditions," and July 26 to 31, considered as "haze conditions." Is there any influence from air mass intrusions? Could there be an impact from pollution transport? I would suggest running back trajectories and exploring this aspect further, as it could be an important contributing factor.

Response: The 72-hour backward trajectories from July 14-19 originated from the East China Sea and passed through the coastal areas of Fujian Province (Fig. R1(a)). In contrast, the 72-hour backward trajectories from July 26-31 originated from the South China Sea and passed through the eastern part of Guangdong Province (Fig. R1(b)). This phenomenon indicates a significant difference in air masses between the two periods. As shown in Fig. R1(c), we can observe that the 72-hour backward trajectories from July 14-19 are associated with clean air masses, suggesting that this airflow primarily plays a role in removing pollutants. Additionally, when considering wind speed and direction, we found that during the clean period, the wind predominantly came from the northeast (Fig. R2(a)). During the haze period, the wind direction was dominated by other directions except for the northeast, particularly the southeast wind (Fig. R2(b)). As shown in Fig. R2(c), when the wind blows from the northeast, the pollutant concentrations are very low, indicating that the northeast wind primarily plays a role in clearing pollutants. However, during other wind directions, the overall $PM_{2.5}$ concentrations were relatively high, with high values mainly occurring at lower wind speeds, suggesting that pollution during the haze period primarily comes from local generation.

[Figure]

Fig. R1 72 h backward trajectories of July 14-19 (a), and July 26-31 (b), and WCWT plots for $PM_{2.5}$ concentrations including July 14-19 and July 26-31 at observation site (c).

[Figure]

Fig. R2 Polar frequency of wind speed and wind direction for clean (a) and haze period (b), polar plot of $PM_{2.5}$ concentrations for whole observation period (c)

I'm not entirely sure that the periods of July 11 and 13 are relevant, as there are peaks exceeding 35 µg m$^{-3}$. I believe a "haze" event should have a consistent prevalence over time (at least for a couple of hours), and not be defined by peak. Please consider this option.

Response: The ambient air quality standards (GB 3095-2012) define that if the 24-hour average value of $PM_{2.5}$ exceeds 35 µg·m$^{-3}$, it surpasses the first-level standard; if it exceeds 75 µg·m$^{-3}$, it surpasses the second-level standard. Xiamen, recognized for its excellent air quality, ranked among China's top 10 cities from 2018 to 2023, achieving 7th in 2018, 4th in both 2019 and 2020, 6th in 2021, 9th in 2022, and returning to 7th in 2023 (mee.gov.cn, last assessed October 30, 2014). In this site, the overall pollutant concentrations are relatively low, with 24-hour average values not exceeding 75 µg·m$^{-3}$. The maximum value was observed on July 30, at only 50.38 µg·m$^{-3}$, indicating that, according to strict standards, there is no haze present. The 24-hour average exceeded 35 µg·m$^{-3}$ on only three days: July 28, 29, and 30. If we use the criterion of having at least two consecutive hours exceeding 35 µg·m$^{-3}$ to make a determination, then only the days from July 26 to 31 meet this requirement. We conducted a correlation analysis between the simulated and observed PAN values and found that the original classification method performed better. The $R^2$ value during the clean period decreased from 0.68 to 0.54, and during the haze period, it decreased from 0.47 to 0.36 (Fig. R3), indicating that the new classification method does not effectively differentiate their patterns. Additionally, when examining the time series of the net generation rate of PAN with and without PAN constraints, we see that this classification method precisely selects the scenario where the net generation rate is negative and exceeds -1 when constrained by PAN (Fig. R4), also suggesting that this classification method better distinguishes between the two different generation mechanisms.

[Figure]

Fig. R3 Correlation between PAN observations and simulated values, where magenta represents the clean period and orange represents the haze period. The figure shows the original classification (a); The figure presents the new classification, with July 26-31 designated as the haze stage and the rest as the clean period (b).

[Figure]

Fig. R4 The time series of PAN net production rate with PAN constrained and without PAN constrained

There is no detailed description of the VOCs measured in the analysis or how they were characterized, and there is no information about the instrument employed. Additionally, there is a lack of information regarding oxidation products and PAN precursors. I believe it is essential to include this information, along with diurnal variability and time series of individual species crucial for PAN formation.

Response: Thank you for pointing this out. The VOC measurements were conducted using a gas chromatography mass spectrometer (GC-FID/MS, TH-300B, Wuhan, China) at an hourly time resolution. This included key VOC species such as alkanes, alkenes, aromatics, and oxygenated VOCs. Detailed information regarding the VOC detection system and calibration procedures is available in our previous study (Liu et al., 2022). We added time series plots of VOCs by category and their daily variations, separately listing the species $C_5H_8$ due to its unique source, which aligns with the subsequent RIR analysis. The concentration of alkanes is the highest, followed by alkenes, OVOCs and aromatics, while halogenated hydrocarbons and $C_5H_8$ exhibit lower concentrations (Fig. R5). Furthermore, VOC concentrations for various species are elevated during haze periods compared to clean periods (Fig. R5). As shown in Fig. R6, the diurnal variation of VOC concentrations for various species are not significant during clean periods, likely due to higher wind speeds that facilitate the dispersion of pollutants. In contrast, during haze periods, the daily variations are evident, with peaks occurring before sunrise, followed by a decline, and then an increase after sunset. This is because the haze period is relatively stable at nighttime, which allows for the accumulation of pollutants, while during the daytime, sunlight converts VOCs into photochemical products like $O_3$ and PAN.

[Figure]

Fig. R5 Time series of VOCs observed at IUE during 10-31 July 2018. The gray shading represents days when the PM$_{2.5}$ hourly daily maximum value exceeded 35 μg·m$^{-3}$.

[Figure]

Fig. R6 The diurnal variations of VOCs during clean (a) and hazy (b) periods. Due to the significantly higher concentrations of alkanes compared to other species, the right vertical axis is used for alkanes, while the remaining VOCs are represented on the left vertical axis.

**Minor comments:**

Abstract: It is not clear at this place of the manuscript where the measurements were conducted.

Response: Thank you for the comment. We have revised the abstract to explicitly state that the measurements were conducted in Xiamen, a coastal city in southeastern China: We addressed the puzzle of summertime PAN formation and its association with aerosol pollution under high O$_3$ conditions in Xiamen, a coastal city in southeastern China, by analyzing continuous high temporal resolution data utilizing box modeling in conjunction with the master chemical mechanism (MCM).

Line 31-32: The abbreviations IOA, RIR, and EKMA are not defined.

Response: Thank you for your valuable feedback. We have updated the manuscript to include definitions for the abbreviations as follows: With an index of agreement (IOA) value of 0.75, the MCM model proves to be an ideal tool for investigating PAN photochemical formation. The model performed better during the clean period (R$^2$: 0.68, slope K: 0.91) than during the haze period (R$^2$: 0.47, slope K: 0.75). Through the machine learning method of XGBoost, we found that the top three factors leading to simulation bias were NH$_3$, NO$_3$, and PM$_{2.5}$. Moreover, the net production rate of PAN becomes negative with PAN constrained, suggesting the existence of an unknown compensatory mechanism. Both relative incremental reactivity (RIR) and empirical kinetic modeling approach (EKMA) analyses indicate that PAN formation in this region is VOC-controlled.

Line 38: PAN was also defined in line 20. Maybe just define the terms once.

Response: Thank you for pointing this out. We have revised the manuscript to define PAN (peroxyacetyl nitrate) only once, at its first mention in line 20, and removed the redundant definition in line 38.

Line 43: VOCs already define in line 21

Response: Thank you for your observation. We have revised the manuscript to remove the redundant definition of VOCs

(volatile organic compounds) in line 43, as it is already defined in line 21. This adjustment helps maintain conciseness in the text.

Line 76: There is no detailed description of the city of Xiamen, and there is no basic meteorological information to provide an idea of the island. I would suggest including general information to get an idea of the location of the study.

Response: Thank you for your valuable suggestion. In the revised manuscript, we have included a detailed description of the city of Xiamen, along with essential meteorological information: Xiamen is one of the fastest urbanizing regions in the southeast China and is also one of the cities with the best air quality in China, where the air quality could represent the future of other Chinese urban regions. Between 2018 and 2023, Xiamen ranked among the top 10 cities in China, achieving positions of 7th in 2018, 4th in both 2019 and 2020, 6th in 2021, 9th in 2022, and returning to 7th in 2023 (mee.gov.cn, last assessed October 30, 2014). Xiamen is located in a low-latitude coastal area, with abundant sunlight and long daylight hours during the summer, resulting in strong solar radiation and rapid photochemical conversion rates. The city is typically influenced by the East Asian monsoon and serves as a transport channel for atmospheric pollutants from both the Yangtze River Delta and Pearl River Delta regions. Additionally, during the summer, Xiamen is often affected by complex meteorological conditions such as typhoons and the West Pacific Subtropical High (WPSH). The WPSH creates weather conditions that promote the formation and accumulation of photochemical pollutants and particulate matter (Wu et al., 2019). This setting provides an ideal "laboratory" for investigating the complexities of summertime PAN formation and its relationship with aerosol pollution under high $O_3$ concentrations. In summer, especially in July, high temperatures, high humidity, and intense radiation are likely to accelerate both the formation and consumption rates of PAN.

Line 78: There are 2 definitions of West Pacific Subtropical High (WPSH) and Western Pacific Subtropical high (WPSH) in line 78 and line 164, respectively. Please define once.

Response: Thank you for noticing this duplication. We have revised the manuscript to define the West Pacific Subtropical High (WPSH) only once, at its first mention in line 78, and removed the redundant definition in line 164.

Line 129. Relative incremental reactivity (RIR) is defined in line 129 but it should be defined in line 31.

Response: Thank you for your feedback. We have moved the definition of relative incremental reactivity (RIR) to line 31, ensuring that it is introduced earlier in the manuscript.

Line 175: Consider homogenizing the way you describe Xiamen, in some parts it is as Xiamen City, Xiamen Island or just Xiamen.

Response: Thank you for your feedback. We have standardized the term "Xiamen City" to "Xiamen." However, we retain the designation "Xiamen Island" to specifically refer to the older urban area, which is more densely populated, has heavier traffic, and is economically more developed. As shown in Fig. R7 (a), Xiamen is located in the southeastern coastal area of China, highlighted in light red, while Xiamen Island, which is part of Xiamen, is situated to the south of the observation site (IUE).

[Figure]

Fig. R7 Location of Xiamen (a), position of IUE in Xiamen (b) and surrounding of IUE (c).

Line 193 Figure 1: Please consider changing the colors of each subplot. The same color for each variable is a bit confusing. Consider including $PM_{2.5}$, in one of the first panels of Figure 1. I consider it important to describe the figure in the main text in order according to the figure itself. It would be easier for the reader.

It is mentioned that the gray area represents the daytime average equal to or greater than 35 ugm$^{-3}$, however, on July 25, the concentration was lower. I would suggest correcting the shaded period. The last period should be from July 26 to 31.

It is mentioned that $O_3$ has a high correlation with PAN ($R^2$: 0.75) however it is noticeable that something happened from July 26 to 31. The $O_3$ concentration increased (around 110 ppb) from July 26 but remained constant from July 28 to 31. The $PM_{2.5}$ has the same behavior but the trend was different in comparison to the other days shaded by gray. Is there any explanation for this? please clarify this part.

Response: Thank you for your valuable feedback. As shown in Fig. R8, we have changed the colors of each subplot to ensure better differentiation between variables and enhance clarity in the presentation. We prioritize meteorological parameters in Figure 1 due to their significant impact on pollutant variations. Additionally, we describe the weather conditions before discussing pollutants, which involves meteorological data. Following this, we include the most important parameters representing haze ($PM_{2.5}$, BC) and photochemical pollution ($O_3$ and PAN). Subsequently, we present HONO, which greatly influences atmospheric oxidation, along with key precursors of $O_3$ and PAN (VOCs and $NO_x$). We believe this order provides a clearer understanding of the interactions.

We meant to indicate that the gray area represents the hourly maximum value exceeding 35 $\mu g \cdot m^{-3}$ rather than the daytime average. When using a daily maximum value greater than 35 $\mu g \cdot m^{-3}$ to define haze, there are 12 days that meet this criterion, as shown in the gray shading area of Fig. R8. When using a daily average value greater than 35 $\mu g \cdot m^{-3}$, only three days meet the requirement: July 28, 29, and 30. A detailed explanation for why July 26 to 31 was not chosen as the haze period is provided in the first question, so it will not be reiterated here.

In the original manuscript, we refer to the correlation coefficient R, not $R^2$. R refers to the correlation between the daily maximum values of $O_3$ and PAN. Considering daily maximum correlations, the correlation between BC and PAN is the strongest, with an R value reaching 0.85, while the correlation between $O_3$ and PAN has an R of only 0.75. Analyzing hourly values reveals that during haze periods, the correlation between PAN and $O_3$ is weaker, with an $R^2$ of 0.47, compared to 0.70

during clean periods. From the 25th to the 31st, a WPSH strengthened and controlled Xiamen, resulting in stable meteorological conditions with light winds (ws=1.04 m·s⁻¹), persistently high temperatures (daily maximum average of 37.82 °C), and high relative humidity (maximum daily average of 81.65 %). These factors created an environment conducive to the accumulation of particulate matter and enhanced the photochemical formation of $O_3$ and PAN. The daily maximum averages of $PM_{2.5}$, $O_3$, and PAN were 49.26 μg·m⁻³, 93.62 ppb, and 1.37 ppb, respectively. The days of the 11th and 21st to 23rd were influenced by the typhoon's periphery, and the 13th also saw the WPSH. Overall, it is evident that when influenced by the WPSH, the trends of $PM_{2.5}$ and $O_3$ are relatively similar, while under the influence of the typhoon's periphery, their trends diverge. We further analyzed the correlation during haze periods based on whether they were influenced by the WPSH, finding a negative correlation between the daily maximum values of $PM_{2.5}$ and $O_3$ during typhoon influence, and a positive correlation when influenced by the WPSH, with the correlation significantly increasing from $R^2$ of 0.14 to 0.71. However, our focus is on the relationship between haze and photochemical pollution, and since the subtropical high is a weather pattern and an external factor, we still choose the daily maximum value being greater than 35 μg·m⁻³ as the criterion for determining haze.

[Figure]

Fig. R8 Time series of trace gases and meteorological parameters observed at IUE during 10-31 July 2018. The gray shading represents days when the $PM_{2.5}$ hourly daily maximum value exceeded 35 μg·m⁻³.

[Figure]

Fig. R9 Correlation between $PM_{2.5}$ and $O_3$ daily maximum concentrations during haze period affected by typhoon's periphery

(a) and by WPSH (b).

Line 228: it mentioned that PAN has unimodal pattern. Where can we observe that? There is no indication of any figure.
Response: Thank you for your observation. We have added a reference to Figure 2 in line 228 to clarify where the unimodal pattern of PAN can be observed.

Line 242: This information is from Mexico as a country or Mexico City? Please clarify it.
Response: Thank you for your question. We have clarified in the manuscript that the information refers specifically to Mexico city, rather than the country of Mexico as a whole.

Line 301: The font and quality of the figures are different compared to the other figures. Please consider increasing the font and resolution.
Response: Thank you for your feedback. We have increased the font size and resolution of Figure 4 to ensure consistency and improve clarity (Fig. R10).

[Figure]

Fig. R10 Feature importance was obtained by XGBoost-SHAP method (a). The scatter plots between concentration of top three important features and their SHAP values (b, c and d), and colored with the bias (the model simulation minus the observed value).

Line 244. According to the main text, PAN undergoes thermal decomposition at high temperatures. However, the specific temperatures are not mentioned, as well as the temperature at 12:00 LT, when the decomposition commonly occurs. Please include this information.
Response: Thank you for your suggestion. The average temperature during the entire observation period was 31.39 °C, with an average temperature of 34.64 °C at 12:00 LT. As shown in Fig. R11, the thermal decomposition of PAN exhibits an exponential relationship with temperature. To highlight the significant impact of temperature, we included the thermal decomposition component in our analysis and found a substantial increase in the slope: from 0.021 during clean periods to 0.1504, and from 0.009 during haze periods to 0.1581 (Fig. R12).

[Figure]

Fig. R11 Correlation between temperature and PAN thermal decomposition during clean (a) and haze (b) period.

[Figure]

Fig. R12. Correlation between PAN and $O_3$ daily maximum concentrations during haze and clean (a), and correlation between PAN+TPAN and $O_3$ daily maximum concentrations during haze and clean (b)

Line 357: The $R^2$ is reported with a different number of decimal places. This $R^2$ has 2 number of decimal places but in line 275 $R^2$ has 4. Please be consistent.

Response: Thank you for bringing this to our attention. We have standardized the reporting of $R^2$ values throughout the manuscript to two decimal places for consistency.

Line 420: AOC is not defined in the main text.

Response: Thank you for pointing this out. We have added a definition for AOC (atmospheric oxidative capacity) in the main text to ensure that readers have a clear understanding of the term.

Line 452: It mentioned that the temperature is a key factor in determining PAN concentrations; however, it does not specify the temperature at which PAN decomposes or if there are chamber experiments that confirm this. It is also important to review laboratory experiments to accurately determine the decomposition conditions.

Response: Thank you for your insightful comments. In response to the previous question regarding temperature (Line 244), we have added detailed information about temperature and demonstrated the exponential relationship between PAN thermal decomposition and temperature (Fig. R11). Additionally, we noted that incorporating thermal decomposition significantly enhances the efficiency of PAN formation (Fig. R12). Regarding laboratory experiments, we reviewed several studies on the thermal decomposition of PAN. For instance, Tuazon et al. (1990) studied PAN in NO-NO$_2$ air mixtures over a temperature range of 283-313 K, yielding a rate constant k = $2.52 \times 10^{16}$ exp (-13573/T) s$^{-1}$. Cox & Roffey (1977) found a rate constant of k =$10^{14.90\pm0.60}$ exp (-104.0 ± 3.2 kJ/RT) s$^{-1}$ over 294-328 K. Senum et al. (1986) reported a rate constant of k = $2.1\times10^{12}$ exp (-24.800 ± 1800 cal/(Kmol/RT)) s$^{-1}$ over 298-338 K. Although there are differences in the coefficients from these studies, they all exhibit an exponential relationship, consistent with our findings. We added the following sentence in the revised manuscript:

Previous laboratory experiments also indicated that the thermal decomposition of PAN is exponentially related to temperature (Cox & Roffey 1977; Senum et al., 1986; Tuazon et al., 1990).

*Reference*

*Tuazon, E. C., Carter, W. P., & Atkinson, R. (1991). Thermal decomposition of peroxyacetyl nitrate and reactions of acetyl peroxy radicals with nitric oxide and nitrogen dioxide over the temperature range 283-313 K. The Journal of Physical Chemistry, 95(6), 2434-2437.*

*Cox, R. A., & Roffey, M. J. (1977). Thermal decomposition of peroxyacetylnitrate in the presence of nitric oxide. Environmental Science & Technology, 11(9), 900-906.*

*Senum, G. I., Fajer, R., & Gaffney, J. S. (1986). Fourier transform infrared spectroscopic study of the thermal stability of peroxyacetyl nitrate. The Journal of Physical Chemistry, 90(1), 152-156.*

Technical corrections

Line 41: Use subscript in NOx like this: $NO_x$

Response: Thank you for your suggestion. We have updated "NOx" to use the correct subscript format.

Line 53: The space between "+" and "products" is missing.

Response: Thank you for your careful review. We have corrected the spacing issue between the "+" and "products" in line 53 to ensure proper formatting.

Line 56: The space between "+" and "$NO_2$" is missing.

Response: Thank you for pointing this out. We have corrected the spacing issue between the "+" and "$NO_2$" in line 56.

Line 80: It is already defined as "$O_3$" however you mention "ozone" (e.g., lines 68, 81, 204, 232, 432, 435, 436, 438, 439) and "$O_3$" (e.g., lines 21, 23, 34, 39, 43, 45, 57, 69, 73, 86) in the text. It should be consistent.

Response: Thank you for your valuable feedback. We have revised the manuscript to ensure consistency in the use of terms. We have used "$O_3$" uniformly throughout the text, aligning with the definitions provided earlier.

Line 93: x in NOx is in italics, but should be in upright font.

Response: Thank you for your observation. We have corrected the font for "x" in NOx to be upright, as suggested.

Figure 2: Please align all legends.

Response: Thank you for your feedback. We have aligned all legends in Figure 2 as requested.

Figure 3: The resolution of the figure needs to improve. The font of the figure does not match the others. Please improve it.

Response: Thank you for your feedback. We have improved the resolution of Figure 3 and ensured the font matches that of the other figures.

Line 306: The figure caption is incomplete. The description of Figure 6b is missing.

Response: Thank you for your observation. We have updated the figure caption to include a complete description of Figure 6b: Feature importance was obtained by XGBoost-SHAP method (a). The scatter plots between concentration of top three important features and their SHAP values (b, c and d), and colored with the bias (the model simulation minus the observed value).

Figure 8: Please change the colors of Figure 8b. It is very confusing with Figure 8a.

Response: Thank you for your feedback regarding Figure 8. As shown in Fig. R13, we have revised the colors in Figure 8b to ensure they are distinct from those in Figure 8a.

[Figure]

Fig. R13 RIR of PAN formation to major precursors (a), the impact of different VOCs species (b)

Figure 10. Please increase the font of the text and the resolution of the Figure.

Response: Thank you for your suggestion. As shown in Fig. R14, we have increased the font size of the text and enhanced the resolution of Figure 10.

[Figure]

Fig. R14 (a) Time series plot of $\Delta$Net (O$_3$) and the reaction of $\Delta$(RO$_2$+NO$_2$), (b) Diurnal variation of $\Delta$Net (O3) during clean and hazy conditions, (c) Correlation between $\Delta$Net (O$_3$) and PAN. $\Delta$Net (O$_3$) is calculated as the base scenario with the PAN mechanism minus the scenario without the PAN mechanism.

---

## Author Response (AR2)

Please find enclosed two referee reports. While one referee is satisfied with your revision, the second has still some issues that have not adequately addressed and need to be corrected.

I also found, while reading your revised manuscript, several issues that should be considered/corrected before publication:

Response: Thank you for providing the referee reports and for your careful review of our revised manuscript. We appreciate the constructive feedback from both referees and yourself, which will undoubtedly improve the quality of our work. We are pleased to learn that one referee is satisfied with our revision. Regarding the concerns raised by the second referee, we have carefully reviewed their comments and provided detailed responses, along with corresponding revisions in the manuscript. Additionally, we have thoroughly addressed the issues you identified during your review. Below, we provide a point-by-point response to your comments, detailing the changes made to the manuscript. We sincerely hope it now meets the journal's standards for publication. Thank you again for your guidance and consideration. We look forward to your feedback.

General comments:

- P6, L177 and throughout the manuscript: It should rather be PM than P.M. Further a consistent way of writing should be used throughout the manuscript, either "PM" in capital letters or "pm" in small letters.

Response: Thank you for pointing this out. We have carefully revised the manuscript to ensure consistency in the notation. All instances of "P.M." have been replaced with "PM," and the usage of "PM" in capital letters has been applied consistently throughout the text.

- Writing of dates: Omit "th" and "rd" (since it is also not done consequently the same way. Sometimes it is written in superscript and sometimes not). Use complete dates instead, e.g. 4 July 2018.

Response: Thank you for pointing this out. We have revised the manuscript to ensure consistency in the writing of dates. We have removed the "th" and "rd" suffixes and adopted the complete date format (e.g., "4 July 2018") throughout the manuscript.

- Writing of Times: It always should be clear if you refer to local times or UTC, so thus either add UTC or LT depend on the correct time.

Response: Thank you for your valuable feedback. We fully agree that it is important to clearly distinguish between local time (LT) and Coordinated Universal Time (UTC). In the revised manuscript, we have ensured that all time references are clearly indicated as either UTC or LT, depending on the context, to avoid any ambiguity.

P1, L26: Add "model" after MCM?

Response: Added.

P3, L89: Same here?

Response: Added.

P5, L147: "in over" should rather be "in" or "over".

Response: Thank you for pointing this out. We have revised the text by removing "over" and keeping only "in" to ensure clarity and correctness.

P6, L153: add "data" so that it reads "test data set".

Response: Added

P6, L166-167: Avoid separation of "Fig." and the number of the figure, "1".

Response: Thank you for your suggestion. We have revised the text to ensure that "Fig." and the corresponding figure number (e.g., "1") are not separated. This adjustment has been applied throughout the manuscript for consistency.

P9, L225-118: Avoid the double closing or even three times closing parenthesis. Add references without parentheses, thus e.g. "(28.0 ppb, Yang et al., 2020)

Response: Thank you for your helpful comment. We have revised the manuscript to avoid the use of double or triple closing

parentheses. Additionally, we have adjusted the reference format as suggested, using a format such as "(28.0 ppb, Yang et al., 2020)" without extra parentheses.

P10, L250-251: Avoid separation of number and unit at the line break.
Response: Thank you for pointing this out. We have corrected the manuscript to ensure that numbers and units are not separated at line breaks. This has been addressed throughout the manuscript to maintain proper formatting.

P10, L252-253: Also here omit the double parenthesis by removing the parentheses around the reference.
Response: Thank you for your suggestion. We have revised the manuscript to remove the double parentheses around the reference, as recommended.

P10, L257: Has the abbreviation "RIR" been introduced?
Response: Yes, the abbreviation "RIR" is introduced for the first time in the "Abstract" section.

P12, L286: captured its formation rate well in general -> captured its formation rate in general well
Response: Revised.

P13, L304: Overestimate what? Be more clear here.
Response: Thank you for your comment. It refers to the overestimation of PAN concentrations. And we have clarified this point in the manuscript: the OBM model tends to overestimate PAN concentration more significantly.

P13, L312: Remove second parentheses around (Fig. S9).
Response: Removed.

P14 and throughout: Use Copernicus style for units. These should be written without a dot in between.
Response: Thank you for your comment. We have revised the manuscript to follow the Copernicus style for units, ensuring they are written without a dot in between. This adjustment has been applied consistently throughout the manuscript, including in the figures, where the units have also been updated accordingly.

P15, Figure 5 caption: Add days or conditions after haze.
Response: Added.

P16, L378 and 379: local time -> LT
Response: Revised.

P16, L382: Have the abbreviations "OVOCs" and "MGLY" been introduced?
Response: Yes, the abbreviations "OVOCs" and "MGLY" are introduced for the first time in the first paragraph of the "Introduction" section.

P17, L405: Had the abbreviation "RIR" been introduced?
Response: Yes, the abbreviation "RIR" is introduced for the first time in the "Abstract" section.

P18, L438: Remove the x between the numbers (check ACP/Copernicus guidelines. I think they use a dot between).
Response: Thank you for your comment. We have revised the text on P18, L438 to replace the "x" between the numbers with a dot, in accordance with the ACP/Copernicus guidelines. Corresponding changes have also been made consistently throughout the manuscript.

P20, L471: cleaning -> clean
Response: Revised.

P20, L474: ppb/ppb -> ppb ppb-1 (with -1 as superscript)

Response: Revised.

P20, L480: increasing -> increase

Response: Revised.

P21, L501 and 502: nos -> No.

Response: Revised.

Referee report

Some of the critical issues were not addressed properly in this round of revision:

1. The VOCs measurements should be described in more detail, which VOCs could be measured by your instrument is not clear. The authors added a figure on VOCs diurnal variations, however, it is not clear which VOCs were included in each category.

Response: Thank you for your comment. In the revised manuscript, we have added Table R1 in supporting information, which provides detailed information on the VOCs species measured and their respective concentrations. This addition clarifies which VOCs are included in each category. We have added the following sentence in the second paragraph of Section 3.1 (Overview of Observation): "Table S1 provides the detailed VOC concentrations observed during the study period."

Table R1 Measured VOC concentrations during 10-31 July 2018 in Xiamen (units: ppt).

| Chemicals | Mean ± SD | Chemicals | Mean ± SD |
|---|---|---|---|
| **Aromatics** | **549±295** | **Alkanes** | **5001±1378** |
| ethylbenzene | 19±15 | ethane | 1315±180 |
| o-xylene | 21±16 | propane | 1059±490 |
| m/p-xylene | 51±39 | isobutane | 415±103 |
| isopropylbenzene | 4±0 | n-butane | 599±142 |
| n-propylbenzene | 6±1 | isopentane | 706±198 |
| m-ethyltoluene | 12±1 | n-pentane | 83±74 |
| p-ethyltoluene | 8±1 | 2,2-dimethylbutane | 4±5 |
| o-ethyltoluene | 7±1 | 2,3-dimethylbutane | 11±19 |
| 1,3,5-trimethylbenzene | 6±1 | 2-methylpentane | 12±16 |
| 1,2,4-trimethylbenzene | 62±7 | 3-methylpentane | 29±27 |
| 1,2,3-trimethylbenzene | 6±1 | n-hexane | 213±110 |
| benzene | 120±59 | 2-methylhexane | 62±12 |
| toluene | 183±168 | cyclohexane | 39±7 |
| styrene | 44±10 | 3-methylhexane | 96±19 |
| **Halocarbons** | **166±172** | n-heptane | 64±14 |
| 1.3-dichloropropene | 33±33 | n-octane | 23±4 |
| trichloroethylene | 2±6 | n-nonane | 13±2 |
| trichloroethane | 67±88 | n-decane | 13±2 |
| tetrachloroethylenez | 4±6 | n-undecane | 25±5 |
| tetrachloroethane | 1±4 | **Alkenes** | **747±337** |
| chloroethane | 59±129 | 1-hexene | 118±48 |
| **OVOCs** | **699±356** | ethene | 161±117 |
| acetone | 369±166 | propene | 135±34 |
| butanone | 266±158 | 1,3-butadiene | 9±17 |
| 4-methyl-2-pentanone | 4±2 | 1-pentene | 1±1 |
| methyl tert-butyl ether | 60±38 | trans-2-pentene | 57±12 |
| **isoprene** | **153±53** | butene | 8±17 |

2. I still feel it is not adequate to make the correlations between BC and PAN daily maximum concentrations, since they occurred at completely different time of day. This might be just another nonsense correlation without any physical and chemical meaning.

Response: Thank you for your insightful comment. To address your concern, we have recalculated the correlations using daily

average concentrations instead of daily maximum values. The results show that both BC and $O_3$ exhibit a strong positive correlation with PAN, with correlation coefficients of 0.77. This consistent finding reinforces the close connection between summertime haze and photochemical pollution observed during the study period. Accordingly, we have made the following revisions in the manuscript: in Section 3.1 (Overview of Observation), the second paragraph now reads: "The correlation between the average daily values of PAN and both BC and $O_3$ is strong, with a correlation coefficient of 0.77 for each (Fig. S6), suggesting that summertime haze and photochemical pollution were deeply connected." Additionally, in the abstract, we have modified it to: "The average daily values of PAN showed a strong correlation with black carbon (BC) (R=0.77) and $O_3$ (R=0.77), suggesting a close connection between summertime haze and photochemical pollution."

[Figure]

Figure R1. The correlation between the average daily values of PAN and BC (a), as well as the correlation between the maximum daily values of PAN and $O_3$ (b).

3. The numbers in NO2, NO3, SO4, CH3CHO etc. in all Figures were not adequately set to subscripts, NO3 should be clarified to stand for nitrate and not NO3 radical to avoid confusion.

Response: We appreciate your attention to detail regarding the chemical notations in the figures. We have corrected all instances where subscripts were not properly formatted. Additionally, we have clarified in the figure captions that $NO_3^-$ refers to nitrate and not the $NO_3$ radical to ensure clarity and avoid confusion. As the figures are numerous, the revisions are reflected directly in the manuscript, and we have not included them here for brevity.